# A 16-year global climate data record of total column water vapour generated from OMI observations in the visible blue spectral range

Christian Borger, Steffen Beirle, and Thomas Wagner

Satellite Remote Sensing Group, Max Planck Institute for Chemistry, Mainz, Germany

**Correspondence:** Christian Borger (christian.borger@mpic.de) and Thomas Wagner (thomas.wagner@mpic.de)

**Abstract.** We present a long-term data set of $1° \times 1°$ monthly mean total column water vapour (TCWV) based on global measurements of the Ozone Monitoring Instrument (OMI) covering the time range from January 2005 to December 2020.

In comparison to the retrieval algorithm of Borger et al. (2020) several modifications and filters have been applied accounting for instrumental issues (such as OMI's "row-anomaly") or the inferior quality of solar reference spectra. For instance, to overcome the problems of low quality reference spectra, the daily solar irradiance spectrum is replaced by an annually varying mean Earthshine radiance obtained in December over Antarctica. For the TCWV data set only measurements are taken into account for which the effective cloud fraction $< 20\%$, the AMF $> 0.1$, the ground pixel is snow- and ice-free, and the OMI row is not affected by the "row-anomaly" over the complete time range of the data set. The individual TCWV measurements are then gridded to a regular $1° \times 1°$ lattice, from which the monthly means are calculated.

The investigation of sampling errors in the OMI TCWV data set shows that these are dominated by the clear-sky bias and cause on average deviations of around -10%, which is consistent with the findings from previous studies. However, the spatiotemporal sampling errors and those due to the row anomaly filter are negligible.

In a comprehensive intercomparison study we demonstrate that the OMI TCWV data set is in good agreement to global reference data sets of ERA5, RSS SSM/I, and CM SAF/CCI TCWV-global (COMBI): over ocean the orthogonal distance regressions indicate slopes close to unity with very small offsets and high coefficients of determination of around 0.96. However, over land, distinctive positive deviations of more than $+10\,\mathrm{kg\,m^{-2}}$ are obtained for high TCWV values. These overestimations are mainly due to extreme overestimation of high TCWV values in the tropics, likely caused by uncertainties in the retrieval input data (surface albedo, cloud information) due to frequent cloud contamination in these regions. Similar results are found from intercomparisons with in situ radiosonde measurements of the IGRA2 data set. Nevertheless, for TCWV values smaller than $25\,\mathrm{kg\,m^{-2}}$, the OMI TCWV data set shows very good agreement with the global reference data sets.

Also, a temporal stability analysis proves that the OMI TCWV data set is consistent with the temporal changes of the reference data sets and shows no significant deviation trends.

Since the TCWV retrieval can be easily applied to further satellite missions, additional TCWV data sets can be created from past missions such as GOME-1 or SCIAMACHY, which under consideration of systematic differences (e.g. due to different observation times) can be combined with the OMI TCWV data set in order to create a data record that would cover a time span from 1995 to the present. Moreover, the TCWV retrieval will also work for all missions dedicated to $NO_2$ in future such as Sentinel-5 on MetOp-SG.

The MPIC OMI total column water vapour (TCWV) climate data record is available at https://doi.org/10.5281/zenodo.7973889 (Borger et al., 2023).

## 1  Introduction

Water vapour is the most important natural greenhouse gas in the Earth's atmosphere altering the Earth's energy balance by playing a dominant role in the atmospheric thermal opacity and having a major amplifying influence on several factors of anthropogenic climate change through various feedback mechanisms (Kiehl and Trenberth, 1997; Randall et al., 2007; Trenberth et al., 2009). Though its great importance not only on processes on global/climate scale, the complex interactions between the components of the hydrological cycle (including water vapour) and the atmosphere are still one of major challenges of climate modelling and for a better understanding of the Earth's climate system in general (Stevens and Bony, 2013). Moreover, the amount and distribution of water vapour are highly variable, so that for global observations these must also be measured with high spatiotemporal resolution. Considering that changes in water vapour are closely linked to changes in temperature via the Clausius-Clapeyron equation, i.e. for typical atmospheric conditions a temperature increase of 1 K yields an increase in the water vapour concentration by approximately 6-7% (Held and Soden, 2000), it is essential to monitor the variability and change of the amount and distribution of water vapour on global scale accurately.

To observe the water vapour distribution on global scale, satellite measurements provide invaluable information. Due to its spectroscopic absorption properties, water vapour can be retrieved from satellite spectra in various different spectral ranges, ranging from the radio (e.g. Kursinski et al., 1997), microwave (e.g. Rosenkranz, 2001), thermal infrared (e.g. Susskind et al., 2003; Schlüssel et al., 2005; Schneider and Hase, 2011), short and near-infrared (e.g. Bennartz and Fischer, 2001; Gao and Kaufman, 2003; Schrijver et al., 2009; Dupuy et al., 2016; Schneider et al., 2020) to the visible spectral range (e.g. Noël et al., 1999; Lang et al., 2003; Wagner et al., 2003; Grossi et al., 2015).

Within the past decade, substantial progress has been made to retrieve total column water vapour (TCWV) within the visible blue spectral range (e.g. Wagner et al., 2013; Wang et al., 2019; Borger et al., 2020; Chan et al., 2020) allowing to make use of measurements from satellite instruments like TROPOMI (Veefkind et al., 2012) and even GOME-2 (Munro et al., 2016) for which so far only retrievals in the visible red and near-infrared spectral range have been available. In comparison to these aforementioned spectral ranges, TCWV retrievals in the visible "blue" have several advatanges, for instance similar sensitivity for the near-surface layers over land and ocean due to a more homogenous surface albedo distribution than at longer wavelengths (Koelemeijer et al., 2003; Wagner et al., 2013; Tilstra et al., 2017). Moreover, any satellite mission dedicated to $NO_2$ monitoring is covering this spectral range.

For investigations of climate change or global warming, respectively, the Ozone Monitoring Instrument (Levelt et al., 2006, 2018) onboard NASA's Aura satellite is particularly interesting: launched in July 2004 it offers an almost continuous measurement data record of more than 16 years up until today. In this study, we make use of OMI's long-term data record and retrieve total column water vapour (TCWV) from its measurements in the visible blue spectral range in order to generate a climate data set.

The paper is structured as follows: in Sect. 2 we describe the data set generation and briefly explain the retrieval methodology and the applied modifications in comparison to the TCWV retrieval from Borger et al. (2020). Then in Sect. 3, we investigate potential sampling errors and how the limitation to clear-sky satellite observations influences the representativeness of the TCWV values of the data set. Furthermore, in Sect. 4 we characterize the data set via an intercomparison to the various global reference TCWV data sets and also to IGRA2 radiosonde observations in Sect. 5. Moreover, we analyze temporal stability of the OMI TCWV data set in Sect. 6. Finally, we briefly summarize our results in Sect. 7 and draw conclusions.

## 2 OMI TCWV data set

### 2.1 Ozone Monitoring Instrument

The Ozone Monitoring Instrument OMI (Levelt et al., 2006, 2018) onboard NASA's Aura satellite is a nadir-looking UV-vis pushbroom spectrometer that measures the Earth's radiance spectrum from 270–500 nm with a spectral resolution of approximately 0.5 nm following a sun-synchronous orbit with an equator crossing time around 13:30 LT. The instrument employs a 2D CCD consisting of 60 across-track rows which in total cover a swath width of approximately 2600 km with a spatial resolution of 24 km × 13 km at nadir increasing to 24 km × 160 km towards the edges of the swath. Launched in July 2004, OMI provides an almost continuous measurement record until today with more than 100000 orbits.

However, since July 2007 OMI has suffered from the so-called "row-anomaly" (RA), a dynamic artefact causing abnormally low radiance readings in the across-track rows, i.e. several rows of the CCD detector receive less light from the Earth, and some other rows appear to receive sunlight scattered off a peeling piece of spacecraft insulation. One plausible explanation for these effects is a partial obscuration of the entrance port by insulating layer material that may have come loose on the outside of the instrument (Schenkeveld et al., 2017; Boersma et al., 2018). Thus, in this study, the affected measurements are excluded for the entire period of the data set.

### 2.2 Methodology and modifications of the spectral analysis

To retrieve total column water vapour (TCWV) from UV-vis spectra from OMI, we apply the TCWV retrieval of Borger et al. (2020) developed for the TROPOspheric Monitoring Instrument (TROPOMI) onboard Sentinel-5P. The retrieval is based on the principles of Different Optical Absorption Spectroscopy (DOAS, Platt and Stutz, 2008) with a fit window between 430–450 nm and consists of the common two-step DOAS approach: first, the absorption along the light path is calculated:

$$\ln\left(\frac{I}{I_0}\right) \approx -\sum_i \sigma_i(\lambda) \cdot \text{SCD}_i + \Psi + \Phi \tag{1}$$

where $I_0$ and $I$ represent the solar irradiance and the radiance backscattered from Earth, respectively, and $i$ denotes the index of a trace gas of interest, $\sigma_i(\lambda)$ its respective molecular absorption cross section, $\text{SCD}_i = \int_s c_i ds$ its concentration integrated along the light path $s$ (the so called slant column density), $\Psi$ summarizing terms accounting for the Ring effect and additional pseudo-absorbers, and $\Phi$ a closure polynomial accounting for Mie and Rayleigh scattering as well as parts of the low-frequency contributions of the trace gas cross sections.

Second, to convert the slant column density to a vertical column density (VCD), we apply the so called airmass factor (AMF):

$$VCD = \frac{SCD}{AMF} \qquad (2)$$

The AMF accounts for the non-trivial effects of atmospheric radiative transfer and depends on the conditions of the retrieval scenario (i.e. aerosol and cloud effects, viewing geometry, and surface properties) as well as the profile shape of the trace gas of interest. The algorithm of Borger et al. (2020) makes use of the relation between the $H_2O$ VCD and the profile shape and iteratively finds the optimal VCD by assuming an exponential water vapour profile shape.

For the application of the algorithm to OMI measurements several modifications had to be applied to the algorithm of Borger

et al. (2020). For climate studies such as trend analyses it is evident to provide a consistent data record. Thus, all rows that have ever been affected by the so called "row-anomaly" are excluded from the data set for the complete time series, which corresponds to approximately half of the OMI swath. Also, instead of a daily solar irradiance an Earthshine radiance is used as reference spectrum within the DOAS analysis. The rationale for using an Earthshine radiance over a solar irradiance is as follows:

– The daily OMI solar irradiance spectra (OML1BIRR version 3) are very noisy and have several gaps causing high $H_2O$ SCD fit errors and thus leading to an overall poor quality of the $H_2O$ VCD data set.

   – By using an annual mean solar irradiance spectrum from the year 2005 (also used during the QA4ECV project; Boersma et al., 2018) a good fit quality can be obtained, however, OMI is also suffering from degradation effects (Schenkeveld et al., 2017). Thus, for the case of climate trend analyses it will be almost impossible to disentangle if a trend signal

originates from the spectral degradation of OMI or indeed from a geophysical trend (see also Fig. A1). By using an Earthshine radiance as reference spectrum these degradation effects will largely cancel out.

   – By using an Earthshine radiance as reference spectrum, also the across-track biases within the OMI swath are strongly reduced (see Panel (c) in Fig. 1) and consequently no destriping is necessary during post-processing (see also Anand et al., 2015).

– However, as a disadvantage of the use of Earthshine spectra, the retrieved $H_2O$ slant columns do not represent absolute slant columns because the Earthshine reference spectra also contain $H_2O$ absorptions. Hence, a slant column representative for the chosen reference sector has to be added to the retrieved values.

For the creation of annual Earthshine reference spectra we selected the Antarctic continent as reference sector (high surface albedo due to snow and ice cover) and the time period of December (i.e. during austral summer) yielding a relatively high

signal-to-noise ratio for our radiance measurements despite large solar zenith angles. Furthermore, only pixels above an altitude of 2000 m above sea level are selected: as the air temperatures are very low there, the water vapour concentrations are very low as well, thus representing a reference atmosphere that is as dry as possible (i.e. the reference SCD or better saying the absolute value of its uncertainty has to be as minimal as possible). Also, to avoid the inclusion of noisy measurements (in

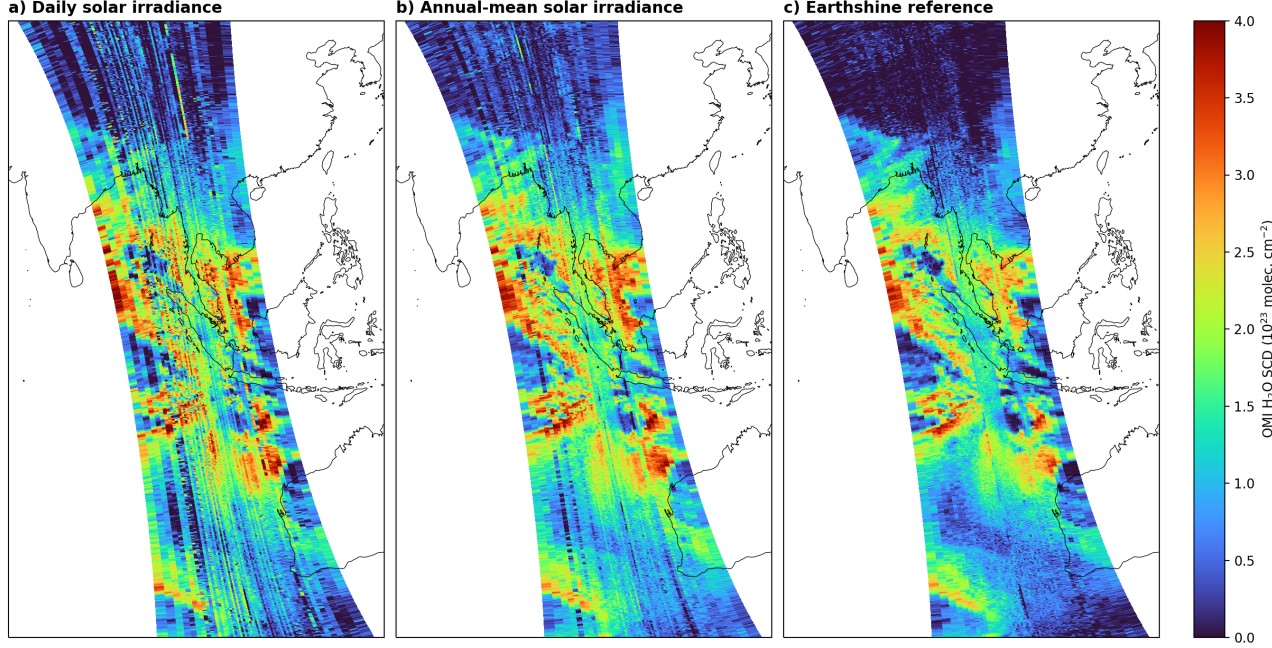

**Figure 1.** Examplary orbit showing the impact of different reference spectrum on the OMI $H_2O$ SCD distribution: a) daily solar irradiance, b) annual-mean solar irradiance, and c) monthly mean Earthshine reference. Orbit 34382, date 01-01-2011.

particular from the descending part of the OMI orbit), only pixels with a solar zenith angle (SZA) < 80° are considered. From these measurements we calculate the monthly-mean radiance for December for each year for every OMI row and then use the resulting reference spectra for the retrievals of the upcoming year.

Figure 1 illustrates the effect of different reference spectra on the $H_2O$ SCD distribution for an examplary orbit. Distinctive stripe patterns are prominent in particular when using the daily solar irradiance as reference spectrum (Panel (a) in Fig. 1). Although the usage of the annual-mean solar irradiance (Panel b) can reduce the strength of the stripes, they are still clearly visible. In contrast, no across-track stripes are detectable for the case of the Earthshine reference and overall the SCDs are also lower due to the $H_2O$ absorption in the Earthshine reference (Panel c).

Further details about destriping in general and a comparison of the temporal behaviour of the irradiance based and Earthshine SCD are available in Appendix A.

## 2.3   VCD conversion and data set generation

To account for the potential water vapour contamination within the Earthshine reference spectra, the SCDs based on the Earthshine reference have to be corrected for the corresponding offset. In this study, we determine this offset $\Delta$SCD for each row based on the difference of the Earthshine based SCDs and solar irradiance based SCDs for the first 5 years of OMI

operation (see Appendix A). Equation (2) can then be rewritten as:

$$\text{VCD} = \frac{\text{eSCD} + \Delta\text{SCD}}{\text{AMF}} \tag{3}$$

where eSCD denotes the SCD derived using the Earthshine reference.

The AMFs are calculated as described in Borger et al. (2020). For the determination of the AMF, additional information about the retrieval scenario like cloud cover and surface properties is necessary. We use the cloud information from the OMI L2 $NO_2$ product (OMNO2, Lamsal et al., 2021) and the modified OMI surface albedo version of Kleipool et al. (2008) as described in Borger et al. (2020). We also tested the surface albedo information from the OMNO2 product, however, within the

framework of a trend analysis study (Borger et al., 2022) we observed spatial artefacts in the surface albedo trends which likely arise from the use of an older version of the MODIS data for the albedo calculation (Lok Lamsal, personal communication). The distribution of TCWV trends is mainly determined by the trends in the SCD. The albedo or AMF trends usually only determine whether the trend signal becomes stronger or weaker, but this only affects trends over land, since an albedo climatology from Kleipool et al. (2008) is used over ocean. As the ice flags from the OMI processor sometimes indicate snow/ice-free surfaces

over Antarctica or Greenland, we additionally use the monthly mean sea ice cover information from ERA5 (Hersbach et al., 2020) and the annual mean land cover information from MODIS Aqua (Sulla-Menashe et al., 2019).

To create the OMI TCWV data set, we have chosen the time range from January 2005 to December 2020 and only include observations with an effective cloud fraction < 20% and AMF > 0.1. Furthermore, the pixels have to be free of snow and ice and must not be affected by the row anomaly. So while about 50% of the orbit is missing because of the RA-filter, the remaining

data still cover an "effective" swath of about 1300 km and is thus still larger than the swaths of GOME-1, SCIAMACHY, or GOME-2A (all about 1300 km) or of the order of SSM/I (about 1394 km). Thus, OMI still achieves complete coverage of the Earth about every 2-3 days, which should provide enough observational data for good representativeness in case of a monthly mean (see also Appendix C and the good agreement to the reference data in Sect. 4). In total, this leaves about 30% of TCWV dat afrom an RA-filtered orbit and about 12% of data from a complete orbit. The results of every orbit are then gridded to a

$1° \times 1°$ lattice for every day. From these daily grids, the monthly mean $H_2O$ VCD distributions are then calculated ensuring that a continuous TCWV time series is available for as many grid cells as possible.

Figure 2 shows the global mean OMI $H_2O$ VCD averaged over the complete time range of the TCWV data set. The resulting distribution demonstrates that the retrieval is capable to capture the macroscale water vapour patterns like high VCD values in the tropics (in particular over the maritime continent) and low values towards the polar regions, but also characteristic regional

patterns like the South Pacific convergence zone.

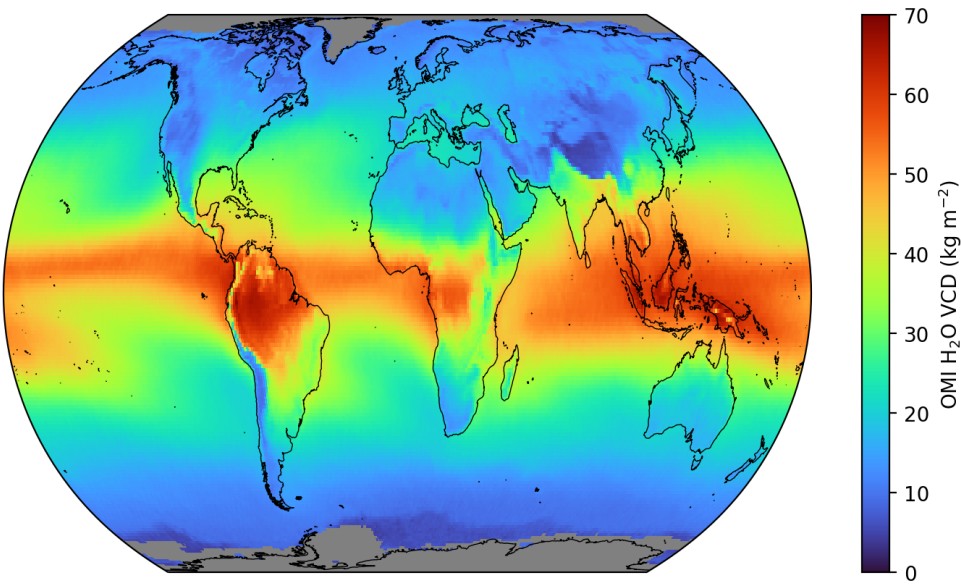

**Figure 2.** Global mean OMI $H_2O$ VCD distribution from 2005 until 2020 based on the OMI analysis using Earthshine reference spectra and corrected for the $H_2O$ SCD bias. Areas with no valid values are coloured grey.

## 3 Sampling errors and clear-sky bias

Although satellite observations enable the analysis of trace gas concentrations on global scale, a fundamental problem is that typically a satellite measurement is only taken once a day for one location. Furthermore, satellite measurements are usually only available under cloud-free conditions, especially in the visible or infrared spectral range and thus no continuous time series is guaranteed. Consequently, they cannot provide a complete picture of geophysical variability, which leads to sampling errors in the calculation of averaged values (e.g. monthly means).

Moreover, the question arises to what extent the limitation to cloud-free pixels influences the monthly averages determined from the OMI satellite measurements, i.e. whether in the OMI TCWV data set a so-called "clear-sky bias" exists. Gaffen and Elliott (1993) investigated this bias using radiosonde ascents and found that the TCWV is about 0-15% lower under cloud-free conditions than under cloudy conditions. Similarly, Sohn and Bennartz (2008) found a clear-sky bias between MERIS and AMSR-E of about 10%.

To estimate the sampling errors, we follow the methods of Xue et al. (2019) and Gleisner et al. (2020): we choose hourly-resolved ERA5 data with a spatial resolution of $0.25° × 0.25°$ as reference data and collocate the ERA5 data with OMI overpass times. These data are then resampled to the $1° × 1°$ resolution of the OMI TCWV data set and the monthly averages are calculated ($\text{TCWV}_{sampled}$). We then take the complete, original ERA5 data, resample it to the same spatial resolution and calculate monthly means from this data as well ($\text{TCWV}_{true}$). The difference between the two data sets then represents the

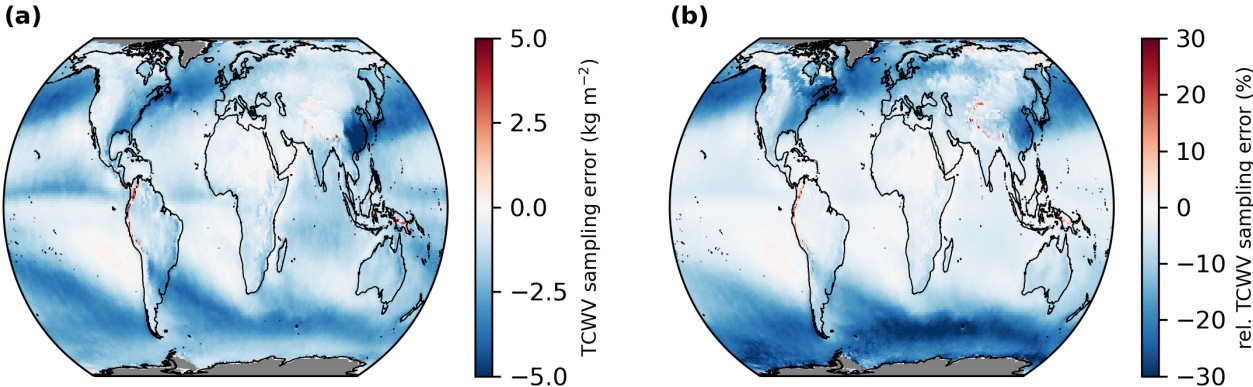

**Figure 3.** Global distributions of the mean sampling errors derived from monthly mean sampling differences for the time range January 2005 to December 2020. Panel (a) depicts absolute sampling error (i.e. $\varepsilon_{sampling}$) and Panel (b) relative sampling error (i.e. $\varepsilon_{sampling}/\text{TCWV}_{true}$). Grid cells for which no data is available are coloured grey.

sampling error:

$$\varepsilon_{sampling} = \text{TCWV}_{sampled} - \text{TCWV}_{true} \tag{4}$$

With this definition, the sampling error summarises the uncertainties due to gaps in the swath, temporal differences or missing data (e.g. due to clouds) (Xue et al., 2019).

Figure 3 shows the mean absolute and relative sampling errors for the complete time range of the OMI TCWV data set (January 2005 to December 2020). Overall, it can be seen that most deviations are negative, i.e. the actual TCWV is underestimated. Regarding the absolute deviations, the strongest deviations can be seen in the area of storm-tracks in the mid-latitudes (e.g. North Atlantic) and the polar regions with values around $-5\,\text{kg}\,\text{m}^{-2}$. The smallest deviations are found in the quasi-permanent cloud-free regions in the subtropics. As expected, the relative differences increase from the equator towards the poles due to the decreasing TCWV values and reach values stronger than -30%.

To investigate to what extent these deviations are related to the clear-sky bias, we proceed similarly to the calculation of the sampling error: we collocate the ERA5 data to the OMI overpass time and once apply a cloud filter (effective cloud fraction < 20%) and once not. Then we resample both data sets to $1° \times 1°$ and calculate monthly means. The difference of both data sets then represents the clear-sky bias:

$$\varepsilon_{clear} = \text{TCWV}_{clear} - \text{TCWV}_{all} \tag{5}$$

To determine seasonal structures, the global distributions of the absolute and relative clear-sky bias for the different seasons were determined from the monthly differences (see Fig. 4). Overall, the distributions of the clear-sky bias correspond very closely to the distributions of the sampling error, both in strength and in pattern. Moreover, the absolute and relative deviations show only slight changes between the different seasons.

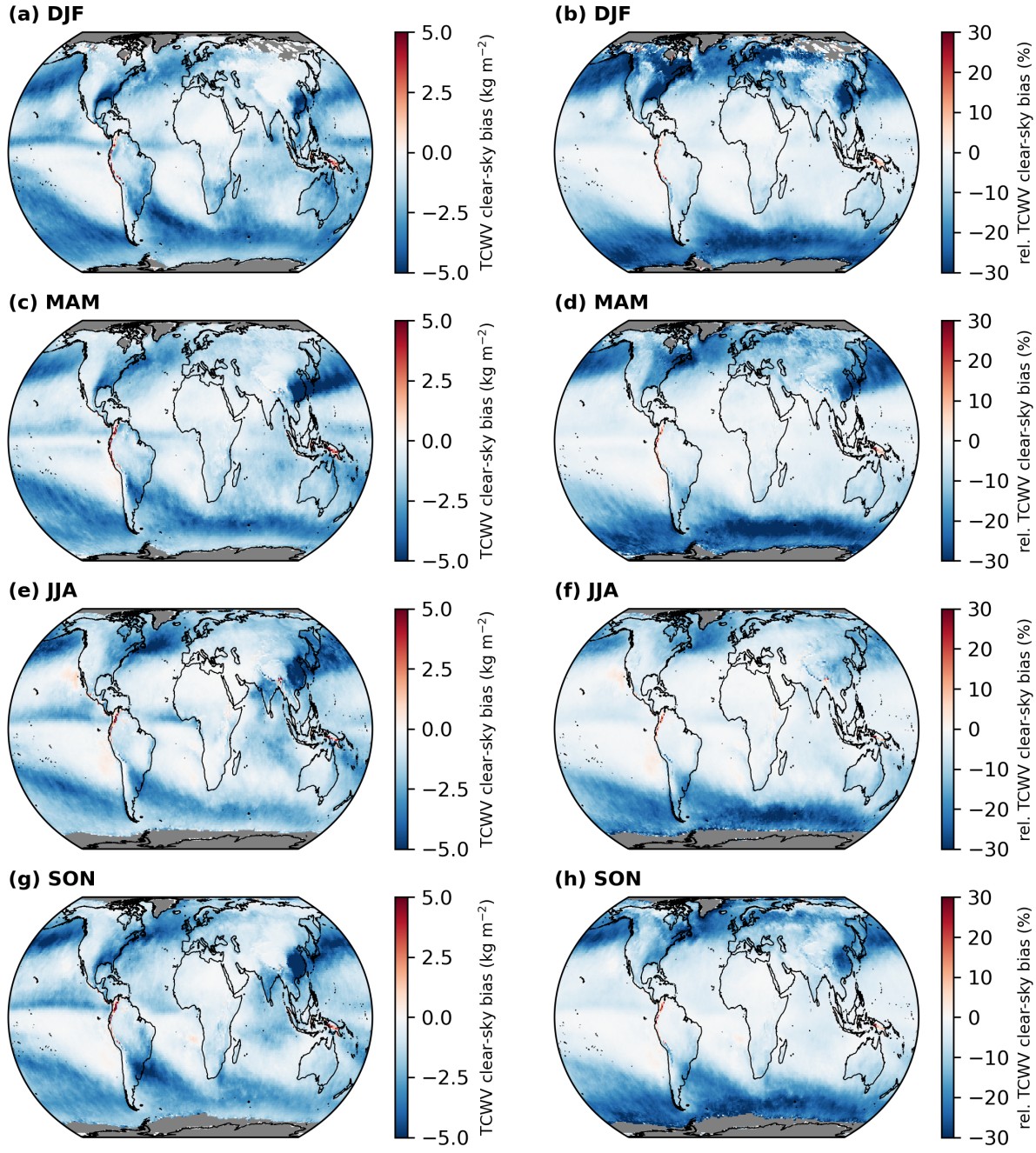

**Figure 4.** Global distributions of the absolute differences ($\varepsilon_{clear}$; left column) and relative differences ($\varepsilon_{clear}$/TCWV$_{all}$; right column) of the mean differences between clear-sky and all-sky ERA5 based on the OMI cloud information for winter (DJF; a & b), spring (MAM, c & d), summer (JJA, e & f), and autumn (SON, g & h) for the time range January 2005 to December 2020. Grid cells for which no data is available are coloured grey

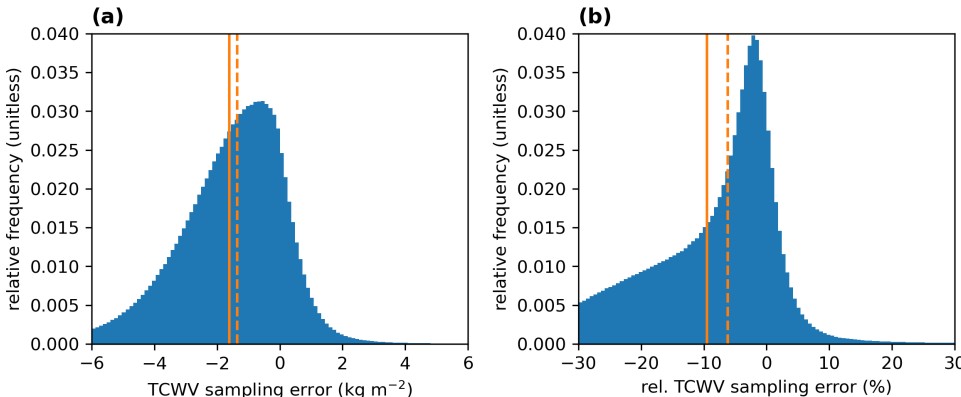

**Figure 5.** Distributions of the absolute differences ($\varepsilon_{sampling}$; Panel a) and relative differences ($\varepsilon_{sampling}/\text{TCWV}_{true}$; Panel b) of the monthly mean differences between clear-sky and all-sky ERA5 data based on the OMI cloud information. The solid and dashed orange line indicate the mean and the median of the distributions, respectively.

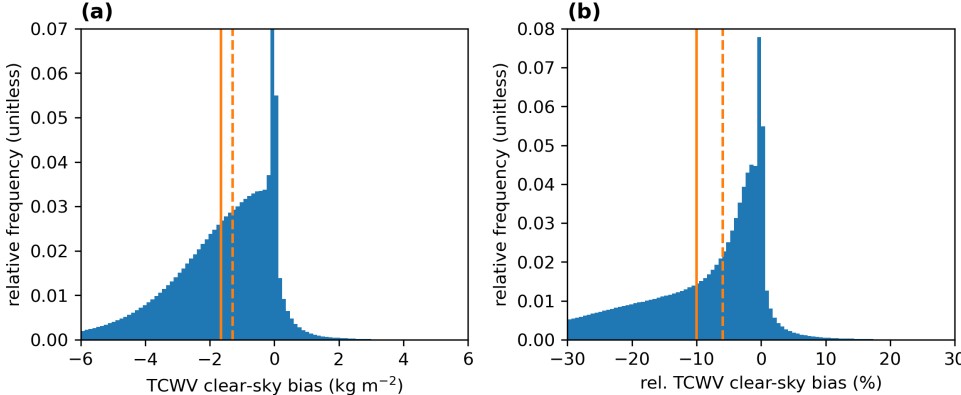

**Figure 6.** Distributions of the absolute differences ($\varepsilon_{clear}$; Panel a) and relative differences ($\varepsilon_{clear}/\text{TCWV}_{all}$; Panel b) of the monthly mean differences between clear-sky and all-sky ERA5 data based on the OMI cloud information. The solid and dashed orange line indicate the mean and the median of the distributions, respectively.

Figures 5 and 6 summarize the sampling error and clear-sky bias distributions, respectively. For the sampling error we obtain a mean absolute deviation of $-1.6\,\mathrm{kg\,m^{-2}}$ (median $-1.4\,\mathrm{kg\,m^{-2}}$) and a mean relative deviation of -9.5% (-6.2%) and for the clear-sky bias we get a mean absolute deviation of $-1.7\,\mathrm{kg\,m^{-2}}$ (median $-1.3\,\mathrm{kg\,m^{-2}}$) and a mean relative deviation of -10.0% (-5.9%). However, the distributions of the absolute and relative deviations for the sampling error and the clear-sky bias are highly left-skewed and thus the mean value in particular is influenced by the long tails of the distributions. Nevertheless, for the clear-sky bias the obtained values agree well with the findings of Gaffen and Elliott (1993) and Sohn and Bennartz (2008).

Since the effect of the clear-sky bias is already included in the sampling error and the results for both errors are very similar, it can be assumed that the spatial and temporal sampling errors play only a minor or negligible role compared to the clear-sky bias.

In addition to the sampling error and the clear-sky bias, we also examined in Appendix C to what extent the monthly means would change if no RA-filter is applied, i.e. if all data of the complete OMI swath were available. It turns out that although deviations arise due to the RA-filter, these deviations are almost an order of magnitude smaller than those of the clear-sky bias and the global distribution of the deviations is mostly noisy. Due to this small influence of the RA-filter, we conclude that the filtered OMI TCWV data are a good representation of the actual TCWV values.

# 4   Intercomparison to existing water vapour climate data records

To evaluate the overall quality of the OMI TCWV data set, we conducted an intercomparison study to various reference data sets of monthly mean TCWV products. For this purpose, we use the merged, 1-degree total precipitable water (TPW) data set version 7 from Remote Sensing Systems (RSS) (Mears et al., 2015; Wentz, 2015), TCWV data from the reanalysis model ERA5 (Hersbach et al., 2019, 2020), and the CM SAF/CCI TCWV-global (COMBI) data set (Schröder et al., 2023) from ESA's Climate Change Initiative (CCI) as reference.

The RSS data set consists of merged geophysical ocean products whereby the values are retrieved from various passive satellite microwave radiometers. These microwave radiometers have been intercalibrated at the brightness temperature level and the ocean products have been produced using a consistent processing methodology for all sensors (more details in Wentz, 2015; Mears et al., 2015). The major advantages of microwave TCVW retrievals are their high precision and accuracy and that they are insensitive to clouds, so that TCWV values can also be retrieved even under cloudy-sky conditions. A disadvantage, however, is that these retrievals are (mostly) only available over the ocean surface.

Thus, we also compare the OMI TCWV data to the CM SAF/CCI TCWV-global (COMBI) data set provided by ESA WV_CCI (Schröder et al., 2023). The CDR combines microwave and near-infrared imager based TCWV over the ice-free ocean as well as over land, coastal ocean and sea-ice, respectively. The data record relies on microwave observations from SSM/I, SSMIS, AMSR-E and TMI, partly based on a fundamental climate data record (Fennig et al., 2020) and on near-infrared observations from MERIS, MODIS-Terra and OLCI (Danne et al., 2022). Hence, it is one of the few (satellite) measurement data sets that provide global coverage over ocean and land surface. Moreover, the data set has been extensively validated

with respect to global reference data sets (e.g. ERA5), satellite products, GPS-measurements from SuomiNet, and GRUAN radiosonde observations (more details in the validation report of Schröder et al. (2023)).

Within comparisons between different satellite data sets a major drawback is the influence of sampling errors due to different observation times, pixel footprint sizes or orbit patterns. To minimise this source of error, data from reanalysis models are useful. ERA5 is the fifth generation ECMWF reanalysis (Hersbach et al., 2020) and combines model data with in situ and remote sensing observations from various different measurement platforms. For our purpose, we use the "monthly averaged reanalysis by hour of day" from the Copernicus Climate Data Store on a $1° \times 1°$ grid. To account for OMI's observation time

(around 13:30 LT), we first calculate the local time for each longitude in the ERA5 data set, then select the TCWV data for the time period between 13:00-14:00 LT and finally merge the selected data.

    For the intercomparison, it is also important to consider that the reference data sets are not perfect or error-free. Thus, we perform an orthogonal distance regression (ODR; Cantrell, 2008).

    In the case of the ODR it is necessary to use reasonable ratios of the relative errors of the compared data sets instead of

using absolute errors in order to obtain meaningful results. In a comprehensive uncertainty analysis, Wentz (1997) determined a typical error of $1.22 \, \mathrm{kg \, m^{-2}}$ for SSM/I observations. Mears et al. (2015) found that the uncertainty of daily microwave TCWV observations for TCWV = $10 \, \mathrm{kg \, m^{-2}}$ was around $1 \, \mathrm{kg \, m^{-2}}$ and for TCWV = $60 \, \mathrm{kg \, m^{-2}}$ around $2$–$4 \, \mathrm{kg \, m^{-2}}$ with respect to GNSS meaurements. Hence, we assume that the uncertainty of the RSS data set is 5% or at least $1 \, \mathrm{kg \, m^{-2}}$. For ERA5 and COMBI we can assume similar uncertainties over ocean, since the TCWV values there are also mainly based on microwave observations.

Unfortunately, no uncertainties are provided for TCWV over land. Thus, for the sake of simplicity, we assume that the relative errors of the reference data sets over land are twice as high as over ocean, i.e. 10% or at least $2 \, \mathrm{kg \, m^{-2}}$. For the OMI TCWV data set we assume an uncertainty of 20% (Borger et al., 2020), but at least $2 \, \mathrm{kg \, m^{-2}}$. We also tested other error assumptions and it turned out that the exact choice of errors is negligible for the regression results as long as the ratio of uncertainties remains similar.

First comparisons with the reference data over land indicated that the OMI data set shows different levels of agreement for low and high TCWV values, with high deviations being particularly prominent for high TCWV values. To be able to estimate the goodness for low and high TCWV, a piecewise linear regression (PWLR) is additionally performed for data over land. For the PWLR, a function of the form

$$f(x) = \begin{cases} a_0 \cdot x + b_0 & x < x_0, \\ a_1 \cdot x + b_1 & x > x_0 \end{cases} \tag{6}$$

is assumed, whereby the function parameters (including $x_0$) are determined via a non-linear least-squares fit.

### 4.1    Intercomparison to RSS SSM/I

The results of the intercomparison between OMI and the RSS TCWV data set are summarized in Figure 7. Figure 7a depicts the 2D histogram from the comparison between the monthly mean values from RSS and the OMI TCWV data set. The data is distributed closely along the 1-to-1 diagonal (black dashed line) and the results of the orthogonal distance regression (ODR, red

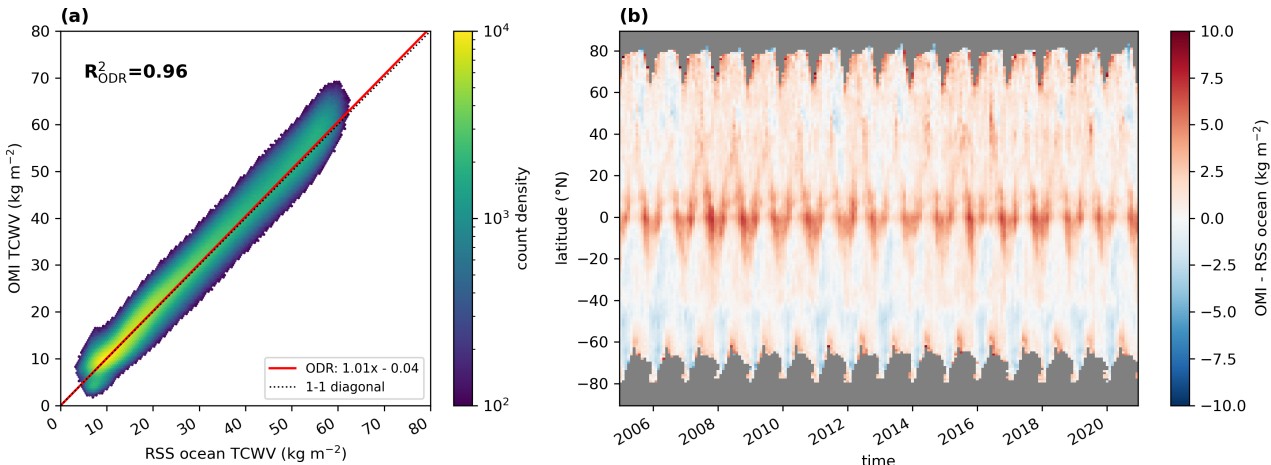

**Figure 7.** Intercomparison between monthly mean TCWV from OMI and Remote Sensing Systems (RSS) merged SSM/I data set for data over ocean. Panel (a) illustrates a 2D histogram in which the colour indicates the count density; the red solid line represents the results of the orthogonal distance regression (ODR). The results of the respective fits are given in the bottom right box and the correlation coefficient in the top left corner. The dashed black line indicates the 1-to-1 diagonal. Panel (b) depicts the TCWV difference of OMI minus RSS within the latitude-time space; reddish colours indicate an overestimation, blueish colours an underestimation of the OMI TCWV data set.

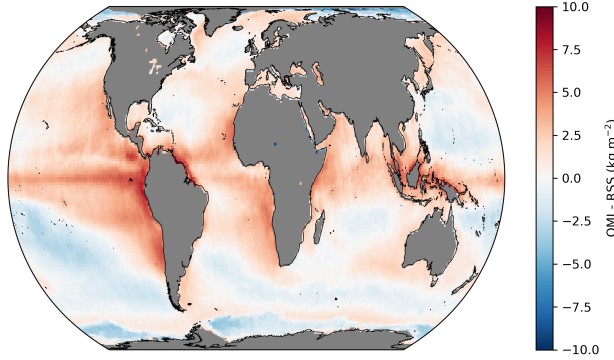

**Figure 8.** Global mean TCWV difference of OMI minus RSS SSM/I for the time range January 2005 until December 2020. Areas with no valid values are coloured grey.

solid line) indicate an overall very good agreement with slopes of around 1.01 and a coefficient of determination of $R^2_{ODR} = 0.96$. If only the TCWV anomalies are compared (i.e. the seasonal cycle is removed), we obtain the correlations of $R^2 = 0.50$.

    Figure 7b illustrates the zonally averaged monthly mean difference of OMI minus RSS TCWV within the latitude-time space. In general, the deviations between OMI and RSS are quite low with a positive bias of $+1.0 \pm 1.5\,\mathrm{kg\,m^{-2}}$. Within the tropics (i.e. between $-20$ to $20\,°\mathrm{N}$) we obtain a mean deviation of $+2.0 \pm 1.6\,\mathrm{kg\,m^{-2}}$ and in the extratropics values of $+0.6 \pm 1.3\,\mathrm{kg\,m^{-2}}$.

However, within the tropics, also distinctive periodic patterns of positive deviations are observable.

Figure 8 shows the global mean TCWV difference between OMI and RSS SSM/I over the complete time period of the OMI TCWV data set. Consistent with the findings from Fig. 7 highest positive deviations can be found in the tropical Pacific ocean and near the coastlines of South America, Africa, and Indonesia whereas strongest negative deviations are obtained around the South Pacific convergence zone and East Siberian Sea. In the case of the tropical Pacific ocean the distribution of the systematic positive deviations matches quite well regions of cold water or of the so called "cold tongue" which is frequently affected by low clouds. Since the highest water vapour concentrations occur in the lower troposphere, small deviations of a few 100 m in cloud top height can have relatively large effects on the AMF (and thus on the retrieved TCWV). In the case of Central America or Atlantic ocean, a too low albedo due to additional absorption by phytoplankton (Kleipool et al., 2008) could explain the systematic positive deviations.

Additional comparisons taking into account only valid grid cells according to the "common-mask" from COMBI data set are presented in Appendix B. This mask filters regions where no continuous time series of data is available or where the data are affected by high uncertainties e.g. due to frequent cloud cover. Therefore only high quality measurements are compared to each other. However, since mainly regions over land surface are affected, the comparisons with the filtered data are almost identical to the unfiltered data.

## 4.2 Intercomparison to ERA5

The results of the intercomparison to ERA5 are depicted in Figure 9. To investigate potential dependencies on the surface type, we separated the data into data over ocean (Fig. 9a & b) and data over land (Fig. 9c & d). The intercomparison for data over ocean reveals similar results as the intercomparison between OMI and RSS: the ODR results indicate a slight overestimation (slopes of around 1.03) together with a coefficient of determination close to unity ($R^2_{ODR}$ of around 0.96). Moreover, the periodic pattern of positive deviations in the tropics occurs again, with an overall small positive bias of $+1.7 \pm 1.7$ kg m$^{-2}$, which increases to $+3.4 \pm 1.7$ kg m$^{-2}$ in the tropics ($-20$ to $20\,°$N) but is around $+1.1 \pm 1.3$ kg m$^{-2}$ in the extratropics. In addition, the correlation of the anomalies is approximately $R^2 = 0.49$.

For data over land, the picture is different: although the ODR gives similar results for the slope as for data over ocean, the distribution in the 2D histogram (Fig. 9c) shows particularly strong positive deviations of approximately $+10$ kg m$^{-2}$ at high TCWV values and an overall systematic offset of around $+1.43$ kg m$^{-2}$. Within the PWLR analysis we find a good agreement to the reference data for TCWV values up to about 25 kg m$^{-2}$ (which represents approximately 74% of all data points) with slopes of around 0.96. However, for higher TCWV values we find distinctive positive overestimations of up to 24%. Nevertheless, even for low TCWV values a systematic offset of approximately $+2.52$ kg m$^{-2}$ is obtained. Also, the correlation of the TCWV anomalies is only around $R^2 = 0.40$.

According to the corresponding latitude-time difference plot (Fig. 9d), the systematic positive deviation in the tropics is now much stronger with values around $+6.2 \pm 3.4$ kg m$^{-2}$ (for latitudes $< 20\,°$), however, in the extratropics the positive deviation is around $+1.7 \pm 1.2$ kg m$^{-2}$ on average and thus of similar magnitude as for the ocean comparisons.

Closer inspection of the mean TCWV difference between OMI and ERA5 (see Fig. 10) reveals that the strong deviations over the tropical landmasses mainly occur in the regions that are affected by frequent cloud cover such as the Amazon basin,

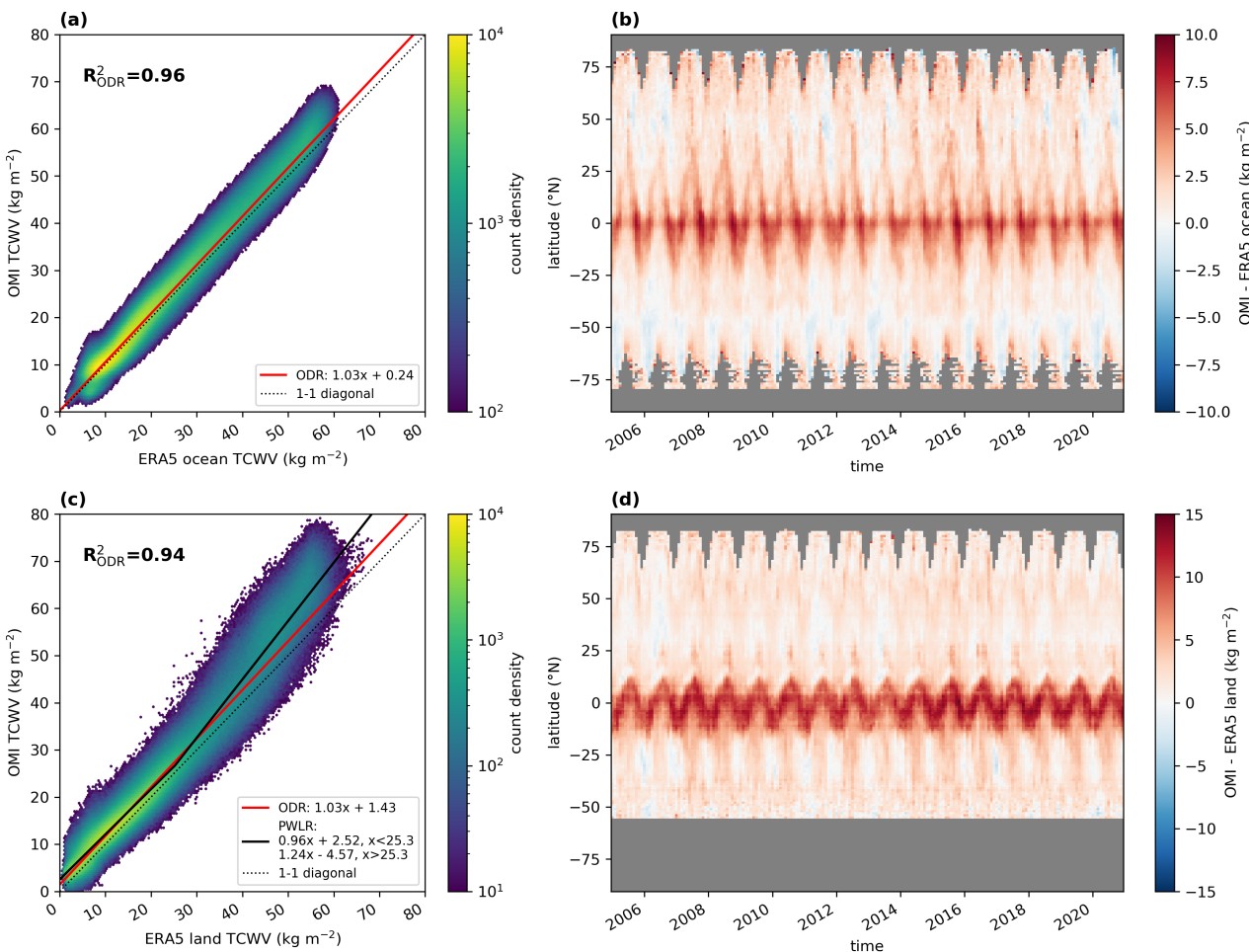

**Figure 9.** Same as Fig. 7, but now with ERA5 data for data over ocean (top row) and for data over land (bottom row). In Panel (c) the red solid line represents the results of the orthogonal distance regression (ODR) and the solid black line the results of the piecewise linear regression (PWLR).

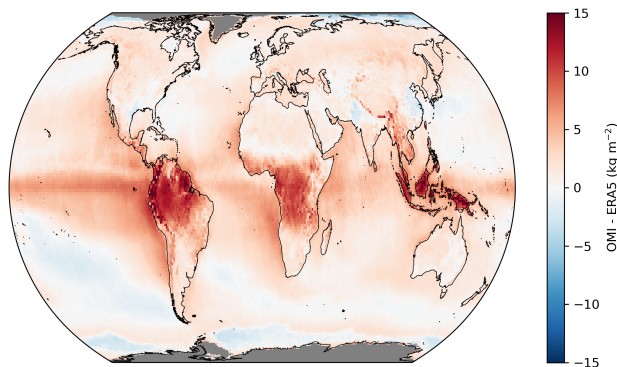

**Figure 10.** Same as Fig.8, but for ERA5.

Central Africa and the maritime continent. In part, these overestimations are further amplified by sampling errors due to complex topography or high mountains (which is also associated with high snow/ice cover and thus fewer valid observations).

Hence, the reasons for the distinctive positive deviations with respect to ERA5 may arise from different causes. For the case of the OMI TCWV retrieval two main uncertainty sources may cause the strong, systematic positive deviations: First, there is the possibility that the used land surface albedo from Borger et al. (2020) is too low, leading to an underestimation of the AMF and consequently to an overestimation of the $H_2O$ VCD. However, Borger et al. (2020) also showed that their modified albedo map led to overall better results for the case of the TROPOMI TCWV retrieval. On the other hand, there may also be uncertainties in the retrieval input data of the cloud information from L2 $NO_2$ product: For example, if the surface albedo is underestimated in the input of the cloud algorithm, this leads to an overestimation of the cloud top height and thus to an underestimation of the AMF, and finally to an overestimation of the $H_2O$ VCD. For the case of ERA5, the frequent cloud cover can be also major source of uncertainty, as only few satellite measurements (or none at all in the thermal infrared) are available due to the frequent cloud contamination. This might lead to clear-sky dry biases in the cloud-affected regions and increased uncertainties within the assimilation process due to the complex radiative transfer in cloudy scenarios (e.g. Li et al., 2016). Likewise, these remote regions are affected by an overall sparseness in the observation density of in situ measurements, so the ERA5 TCWV values are likely to be based mainly on modelled data. Overall, the strong positive deviation of the OMI TCWV data set thus likely results from a combination of an overestimation of the OMI TCWV retrieval and an underestimation of the ERA5 data.

One way to address these errors would be to develop an independent albedo and cloud product, but this is far beyond the scope of this paper. Moreover, under consideration of the soon demise of OMI (probably in 1-2 years) and the ongoing reprocessing of L1 data such an algorithm development would not be worthwhile at the moment of writing this paper.

Hence, considering these large uncertainties in the OMI retrieval and that the uncertainties in ERA5 for data over tropical landmasses are not negligible anymore, we conclude that the OMI TCWV data set can well represent the global distribution of

the atmospheric water vapour content at least over ocean. Over land, however, the data set should be treated with caution due to the systematic positive deviations from the reference data sets, especially in areas of high TCWV values (i.e. above 25 kg m$^{-2}$).

An additional comparison in which particularly critical regions were filtered using the "common mask" from the COMBI data set (see Fig. B1) is given in the Appendix B. When this mask is applied, only high quality measurements are taken into account for the intercomparison. As a result, the extreme overestimations are filtered out and the distribution in the 2D histogram for the comparison over land improves considerably (see Fig. B6a). The slope of the ODR is now around 0.96, which is closer to the results of the PWLR regression for TCWV < 25 kg m$^{-2}$.

## 4.3 Intercomparison to COMBI

For the intercomparison with the COMBI data set we resampled the CDR from its native spatial resolution ($0.5° \times 0.5°$) to the lattice of the OMI TCWV data set. Furthermore, though COMBI covers a time span from July 2002 to December 2017, we focus on the time period January 2005 to March 2016, as the CDR's difference relative to ERA5 over land is only stable over the MERIS and MODIS period, i.e., from 2002 until March 2016 if looking at clear-sky data. For the sake of completeness, the results for the comparison over the complete time range are depicted in the Appendix-Figures E1 and E2.

Figure 11 summarizes the results of the intercomparison. Not surprisingly, the results for data over ocean (Fig. 11a) are similar to the findings of the RSS SSM/I and ERA5 comparison as measurements from the same (or similar) sensors have been considered: the ODR results indicate indicate slight overestimations of around 2% with a coefficient of determination of around 0.95 and the time-latitude diagram indicates an average deviation of +1.3±1.7 kg m$^{-2}$ (+2.5±2.0 kg m$^{-2}$ in tropics, +0.8±1.4 kg m$^{-2}$ in extratropics). However, the correlation of the TCWV anomalies is slightly lower in comparison to the other data sets with values around R$^2$ =0.45 over ocean.

Similar to the intercomparison of ERA5, the intercomparison over land (Fig. 11c) shows roughly similar ODR fit results as over ocean, but here we also find striking positive deviations for high TCWV values and an overall positive offset of 2.23 kg m$^{-2}$. Again, when applying a piecewise linear regression analysis we obtain good agreement with slopes of around 0.95 for TCWV values to about 25 kg m$^{-2}$ but still a distinctive positive offset of 3.51 kg m$^{-2}$ for low TCWV values and distinctive overestimations of up to 33% for higher TCWV values, which is even higher than for the comparison to ERA5. Consequently, the systematic deviations are also much stronger (see. Fig. 11d) and reach values of around +7.2±3.6 kg m$^{-2}$ in the tropics, around +2.7±1.4 kg m$^{-2}$ in the extratropics, and a global average of +4.1±3.1 kg m$^{-2}$. These even higher deviations compared to the analysis with ERA5 could be due to the different observation times of the data sets: MERIS on Envisat and MODIS on Terra have an overpass time of 10:00 LT and 10:30 LT, respectively, and follow a descending orbit, whereas OMI measures at 13:30 LT in an ascending orbit. This might also explain the worse correlation of anomalies of R$^2$ =0.32 for data over land.

Overall, similar to the comparison to ERA5 the strongest positive deviations occur again over the tropical landmasses that are mostly affected by frequent cloud cover (see Fig. 12). Likewise, further overestimations appear in areas with complex high topography (e.g. Indonesia, Andes, Himalayas), suggesting sampling errors when merging the spatial resolutions of the data sets and missing observations due to snow/ice cover.

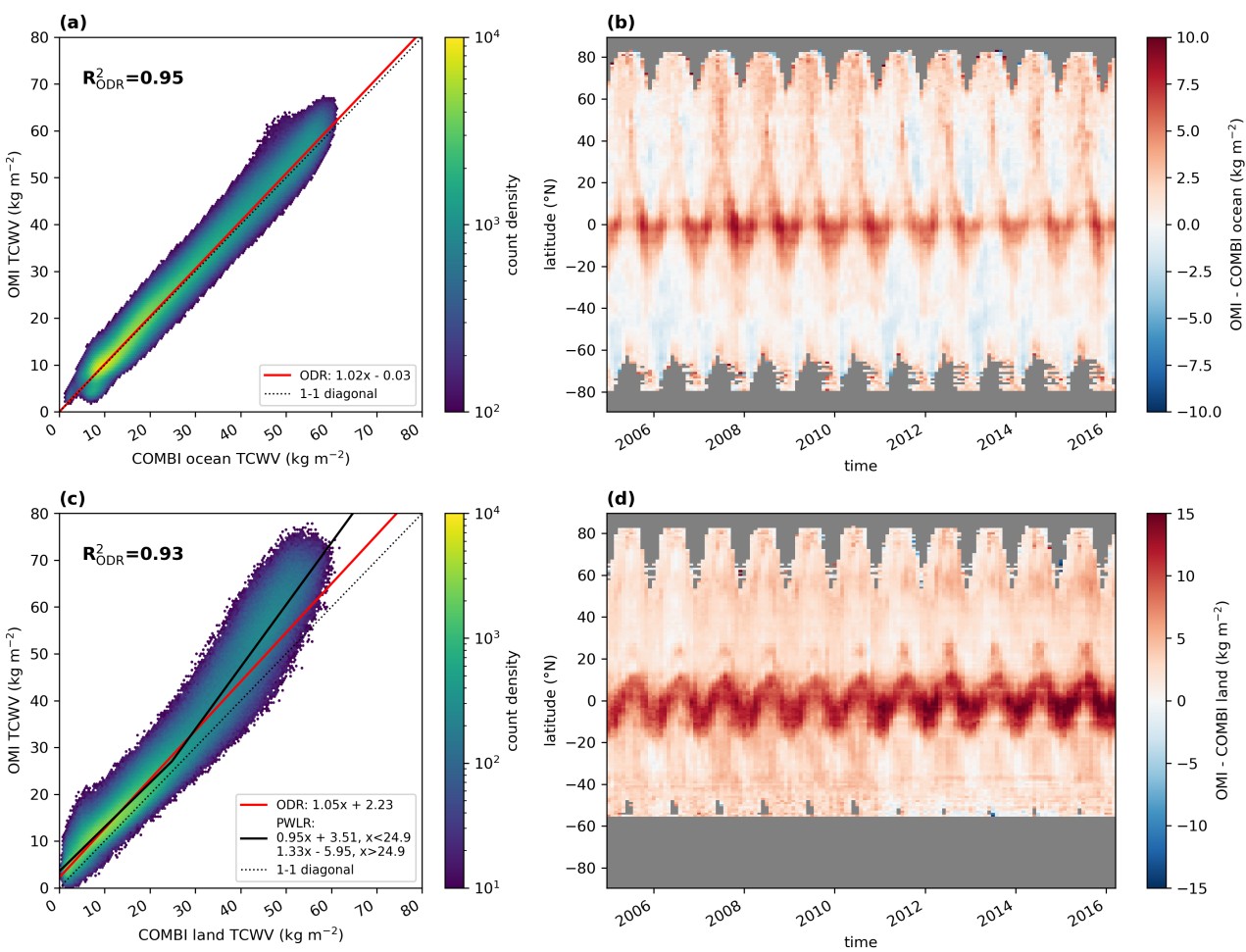

**Figure 11.** Same as Fig. 7, but now with COMBI data for data over ocean (top row) and for data over land (bottom row). In Panel (c) the red solid line represents the results of the ODR and the solid black line the results of PWLR.

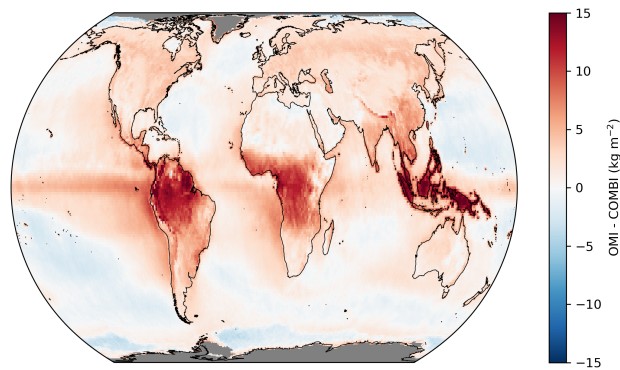

**Figure 12.** Same as Fig.8, but for COMBI.

In Appendix B we present a comparison in which critical regions were filtered using the "common mask" from the COMBI data record. When this mask is applied, there are clear improvements for the comparison over land: the prominent overestimates at high TCWV values are filtered out and the distribution is now closer to the 1-1 diagonal (see Fig. B6b). For the ODR, the slope is around 0.98, which agrees quite well with the slopes obtained for the piecewise linear regression for TCWV <
$25 \, \mathrm{kg \, m^{-2}}$.

## 5 Intercomparison to IGRA2 radiosonde observations

For further comparisons besides reanalysis and satellite data, in situ measurements from radiosondes are invaluable, as these measurements can provide information on the vertical water vapour distribution with high accuracy (Dirksen et al., 2014). Here, the Integrated Global Radiosonde Archive (IGRA) is particularly well-suited for global intercomparisons: IGRA is a
collection of historical and near-real-time radiosonde and pilot balloon observations from around the globe (Durre et al., 2006, 2018) provided by the National Centers for Environmental Information (NCEI) of the National Oceanic and Atmospheric Administration (NOAA). For IGRA version 2 (IGRA2; Durre et al., 2016, 2018), 40 data sources were converted into a common data format and merged into one coherent data set which then went through a quality-assurance system. While to our knowledge no explicit uncertainty estimates have been conducted for water vapour measurements, the IGRA2 humidity measurements are
subject to rigorous quality control (Durre et al., 2018) and the completeness of the IGRA2 humidity observations has also been checked by Ferreira et al. (2019).

Although IGRA2 also provides TCWV data, these are calculated from the surface only up to 500 hPa. Typically, this pressure level is at about 5 km above mean sea level, so if one assumes a typical scale height of the water vapour of 2.1 km (Weaver and Ramanathan, 1995), a low-bias of 10% could be introduced. Thus, to ensure a consistent calculation of the TCWV monthly
means from the IGRA2 data, the following criteria were applied to the individual radiosonde ascents:

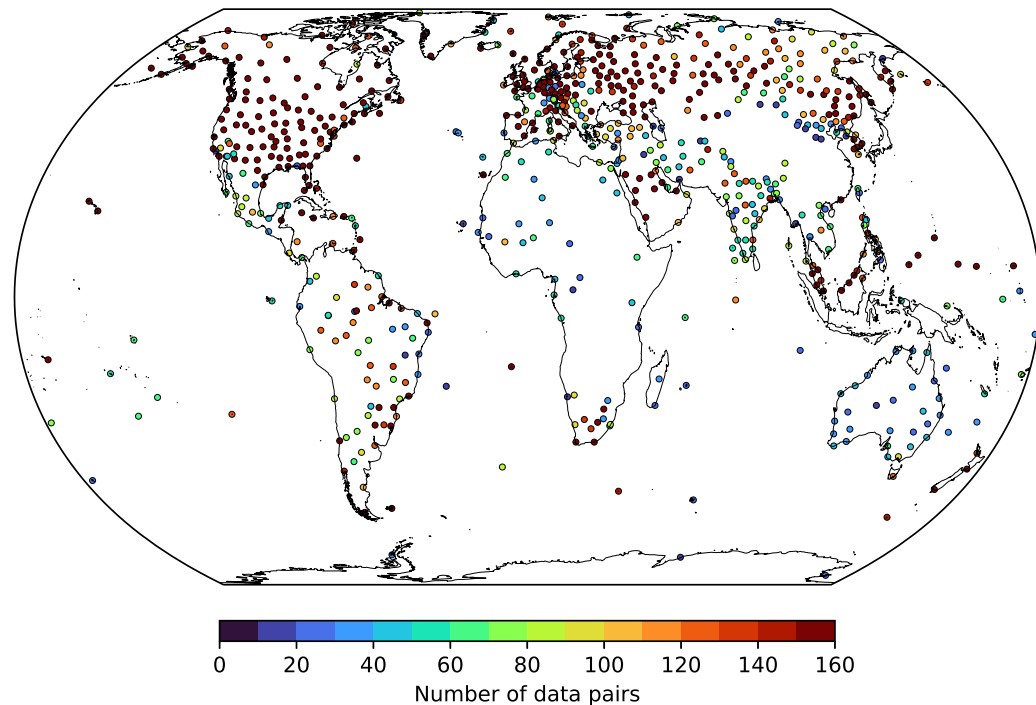

**Figure 13.** Global distribution of radiosonde stations used for the TCWV intercomparison to the MPIC OMI data set. Colours indicate the amount of data pairs for the intercomparison.

1. Only radiosonde ascents that have reached an altitude of at least 300 hPa were considered for the calculation of the TCWV. This pressure level corresponds to a typical geometric altitude of around 9 km. This ensures that the radiosondes covered a large part of the troposphere and thus captured the majority of the TCWV without introducing non-negligible low biases.

2. For the calculation of the monthly means, valid radiosonde ascents of at least 10 different days in the month must have taken place in order to achieve a good temporal coverage of the month.

3. Only stations with at least 12 valid data pairs between the monthly means of IGRA2 and MPIC OMI were considered for the statistical analysis.

Figure 13 shows the global distribution of the locations of the radio sounding stations as well as the numbers of valid
data pairs used for the comparison. Altogether, 731 different radiosonde stations are considered for this comparison study. In addition to a high density of measurement stations, there is a general good temporal coverage in the northern mid latitudes (especially in North America and Europe) and thus good temporal collocation between MPIC OMI and IGRA2 data. For the other parts of the world, however, the measurement network is much less dense and hence the number of temporal collocations

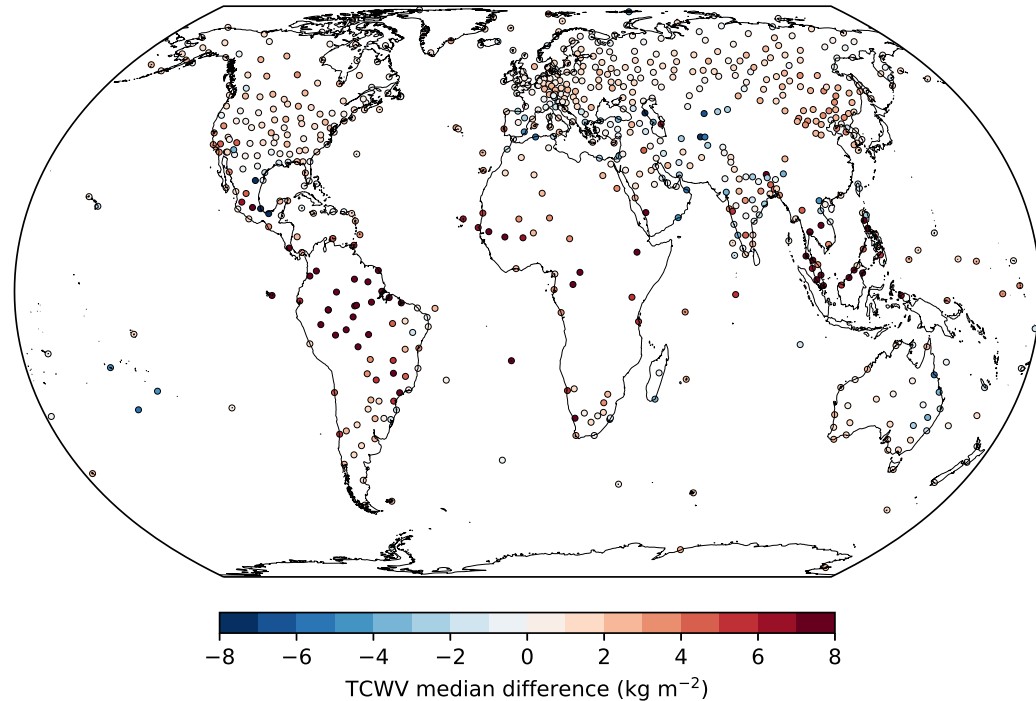

**Figure 14.** Global distribution of the median TCWV difference between the monthly means of the MPIC OMI TCWV data set and those derived from the IGRA2 radiosoundings.

of the two data sets do not reach the values from the Northern mid latitudes. Thus, due to the limited sample size at many
stations the median of the deviation is now used instead of the mean deviation for the comparisons.

The distribution of these median deviations is given in Fig. 14. Overall, the results are consistent with the findings from the previous comparisons with the global satellite and reanalysis data sets (see Sect. 4), in which a good to very good agreement was found for extratropics and an overestimation for the tropics. On average, the median deviation is about $+1.6\pm3.4\,\mathrm{kg\,m^{-2}}$, with about $+0.9\pm2.0\,\mathrm{kg\,m^{-2}}$ in the extratropics and $+4.3\pm5.5\,\mathrm{kg\,m^{-2}}$ in the tropics.

Nevertheless, this comparison to radiosonde measurements demonstrates that the MPIC OMI TCWV data set is also in good to very good agreement to in situ reference data sets, but tends to be systematically overestimated in the tropical landmass regions which is in line with the previous findings from the comparisons with reanalysis and satellite data (Sect. 4).

## 6 Temporal stability

In addition to a good agreement to existing reference data sets, the temporal stability is an important property of a climate
data record. As the COMBI data set only covers the time range up to December 2017, we focus on the comparison to the RSS

SSM/I and ERA5 data sets as these two cover the complete time range of OMI TCWV data set. For the sake of completeness, however, we also show the results for COMBI.

To assess the stability of the OMI TCWV data set, first the global mean relative deviation $\langle \epsilon \rangle$ is derived for every time step:

$$\langle \epsilon \rangle = \frac{\langle \mathrm{OMI} - \mathrm{TCWV}_{ref} \rangle}{\langle \mathrm{TCWV}_{ref} \rangle} \tag{7}$$

For the calculation of global means only data points or grid cells are taken into account for which for every time step data from the OMI TCWV and reference data set are available. In the case of the COMBI data set a "common mask" has been provided (see also Fig. B1).

Then, temporal linear trends of these deviations are calculated using a generalized least-squares (GLS) regression for the fit function:

$Y_t = m + b \cdot X_t = \mathbf{M}x + N_t \tag{8}$

with the intercept $m$, the trend $b$, the increasing time index $X_t$ (in months), which can all be summarised in a matrix $\mathbf{M}_t$. The term $N_t$ stands for the fit residuals with respect to the time series. To account for the temporal autocorrelation of the fit residuals $N_t$ of the GLS, the Prais-Winsten transformation (Prais and Winsten, 1954; Greene, 2019) is used assuming that the residuals follow an autoregressive process. For this purpose, the autocorrelation (ACF) is estimated using the Gaussian-kernel-based 420   cross-correlation function algorithm, as described in Rehfeld et al. (2011). For the estimation of the order of the AR model, we use the partial autocorrelation function (PACF) and investigate after which lag all values of the PACF lie within a confidence interval $\pm\delta$. Assuming that the PACF values for high lags follow a white noise, the confidence interval is defined by the Z-score (in our case of a significance level of 95%) and the length of the time series $L$ according to the following formula (Box et al., 2015):

$\delta = \dfrac{Z}{\sqrt{L}} \tag{9}$

An AR model can then be created from the determined AR order, which is then used to transform the GLS using the transformation matrix $\mathbf{P}$:

$$\mathbf{P}Y_t = Y_t' = \mathbf{P}(\mathbf{M}_t x + N_t) = \mathbf{M'_t}x + \varepsilon_t \tag{10}$$

For details about the construction of the transformation matrix we refer to Weatherhead et al. (1998), Mieruch et al. (2008), 430   and Borger et al. (2022). The trends are then determined from the transformed system in Eq. (10) by simple linear algebra. The results and their uncertainties then already include the effect of the temporal autocorrelation.

Figure 15 illustrates the temporal variability of the relative differences of the OMI TCWV data set and RSS SSM/I, ERA5, and COMBI for the time range January 2005 to March 2016 (blue dashed lines) and January 2005 to the end of the respective data set (blue solid lines). For all three data sets and all time ranges, the PACF analyses showed that an AR(3) model is the 435   most appropriate choice. For the time series until March 2016 we find trends of $+0.21 \pm 1.20$ % per decade for the comparison

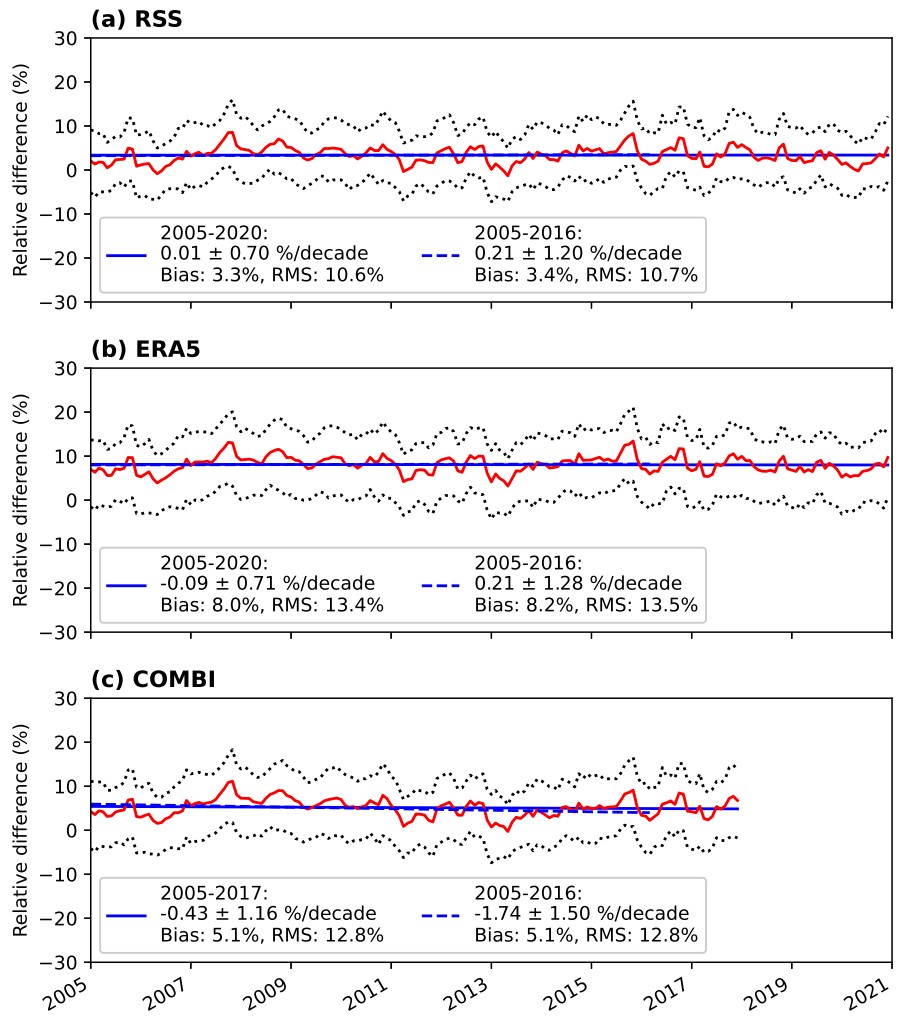

**Figure 15.** Stability analyses of the global mean relative deviations of the OMI TCWV data set with respect to **(a)** RSS SSM/I, **(b)** ERA5, and **(c)** COMBI. Red line: global mean relative deviation; blue line: results of the transformed GLS regression; dotted black line: 25th and 75th percentiles, respectively. Dashed lines represent data for the time range from January 2005 to March 2016 and solid lines represent data for the time range from January 2005 to the end of the respective data set. The bias and RMS provided in the legends correspond to the time seriues of the global mean deviation for the respective time range.

to RSS SSM/I, $+0.21 \pm 1.28\,\%$ per decade for the comparison to ERA5, and $-1.74 \pm 1.50\,\%$ per decade for the comparison to the COMBI data.

For the time series until the end of the reference data set one finds trends of $+0.01 \pm 0.70\,\%$ per decade for the comparison to RSS SSM/I and $-0.09 \pm 0.71\,\%$ per decade for the comparison to ERA5. Moreover, the statistical analyses reveal that these trends are not significantly different from $0\,\%$ per decade. For the comparison to the COMBI data there is a stronger trend (around $-0.43 \pm 1.16\,\%$ per decade) than for the other two data sets, however also the time range is much shorter and does not cover the complete time range of the OMI TCWV data set. Altogether, the obtained trends of the relative deviations are in line with typical stability requirements for climate data products of $\pm 1\,\%$ per decade (see e.g. Beirle et al. (2018) and references therein or the ESA WV_cci user requirements; https://climate.esa.int/media/documents/Water_Vapour_cci_D1.1_URD_v3.0. pdf; last access: 23 May 2023). Moreover, these trends are also in line with the recently published stability requirements for Essential Climate Variables (ECV) according to the Global Climate Observing System (GCOS) implementation plan with stabilities of $\pm 0.1\,\%$ per decade as "goal", $\pm 0.2\,\%$ per decade as "breakthrough", and $\pm 0.5\,\%$ per decade as "target" stability (see GCOS-245; https://library.wmo.int/doc_num.php?explnum_id=11318; last access: 23 May 2023).

To understand to what extent the temporal stability differs over land and over ocean, the data were separated and analysed. The results of this separate analyses are shown in Fig. 16 (over ocean) and Fig. 17 (over land). The RSS data set was not investigated again as it is only available over ocean and therefore redundant to re-examine. The PACF analyses revealed that for all stability analyses over ocean a AR(3) model and over land a AR(2) model are the most appropriate choices.

Over ocean, the OMI data set also meets the $1\,\%$ per decade stability criterion (and also various GCOS stability criteria) for both the long and short periods for the case with ERA5 as reference ($+0.01 \pm 1.17\,\%$ and $-0.28 \pm 0.67\,\%$ per decade, respectively). In contrast, no stability criterion for the comparison with the COMBI data set is fulfilled for both time periods any more ($-0.87 \pm 1.08\,\%$ per decade for the longer and $-1.78 \pm 1.39\,\%$ per decade for the shorter time period). This is surprising, since both reference data sets should consist largely of similar measurement data from mainly microwave satellites. ERA5 is possibly better constrained again due to its larger volume of assimilated observation data.

Over land, the situation is even more complicated: while for ERA5 the $1\,\%$ stability criterion is still met at $+0.62 \pm 0.96\,\%$ per decade for the period from 2005 to 2020, this is no longer the case for the shorter period at $+1.24 \pm 1.75\,\%$ per decade. In the case of the COMBI data set, the stability criterion is not even close to being fulfilled either for the period 2005-2017 ($+3.36 \pm 2.04\,\%$ per decade) or for the period 2005-2016 ($-0.79 \pm 2.39\,\%$ per decade).

Considering the obtained results, it seems that both stability trends over land and ocean largely cancel each other. However, one reason for the high relative deviations over land could be that mainly desert-like regions are used in the analysis due to the aforementioned filter criterion. Thus, rather low TCWV values are used in the normalisation, which means that extreme relative deviations can occur even with rather small, absolute deviations.

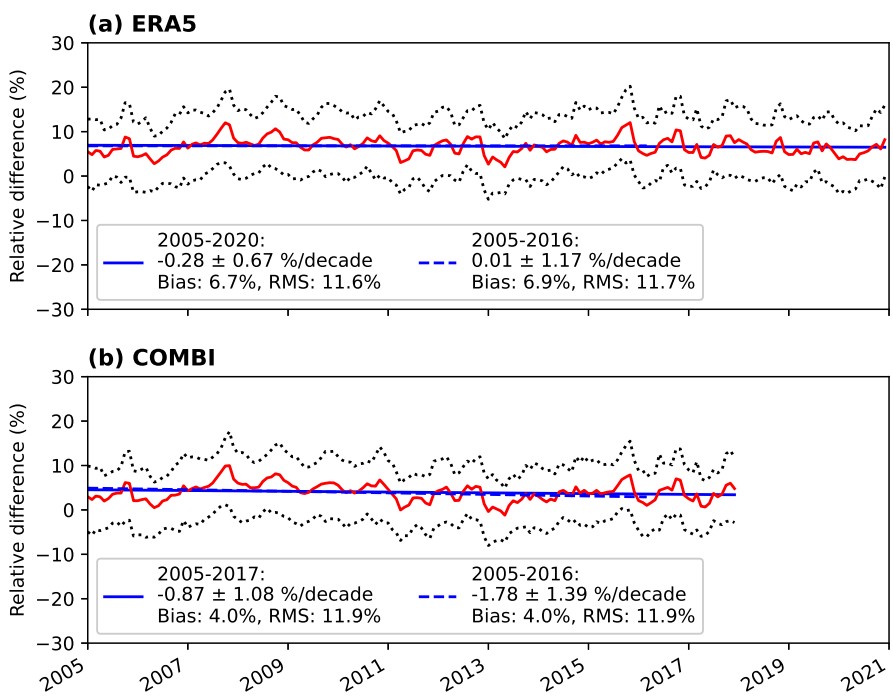

**Figure 16.** Same as Fig. 15, but only for (**a**) ERA5 and (**b**) COMBI and only for data over ocean.

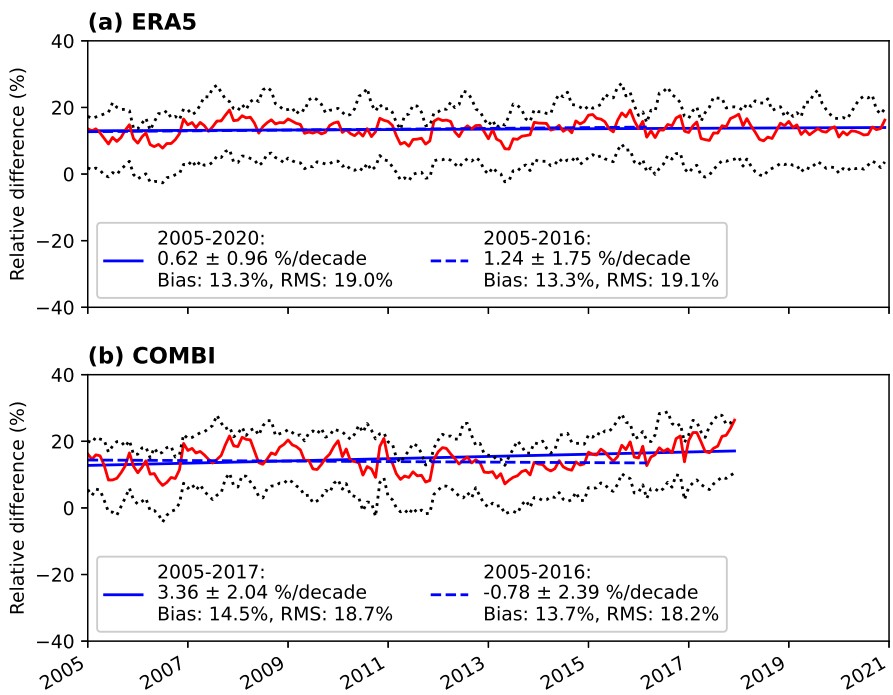

**Figure 17.** Same as Fig. 15, but only for (**a**) ERA5 and (**b**) COMBI and only for data over land.

## 7 Summary

In this study, we present a long-term 16-year data record of total column water vapour (TCWV) retrieved from multiple years of OMI observations in the visible blue spectral range by means of Differential Optical Absorption Spectroscopy. To derive TCWV from OMI measurements, we applied the TCWV retrieval developed for TROPOMI (Borger et al., 2020) and modified the spectral analysis to account for the degradation of OMI's daily solar irradiance. Thus, annual Earthshine reference spectra were calculated from radiance measurements over Antarctica during December (austral summer).

The estimation of the sampling errors in the OMI TCWV data set results in average errors of about -10% (and -6% for the median) and that the largest deviations occur mainly in the the mid-latitude storm tracks and polar regions. Further investigations show that the large deviations of the sampling error correlate well with the deviations of the clear-sky bias. However, the investigation of a seasonal effect of the clear-sky bias did not show any seasonal dependence. Considering the dominant role of the clear-sky bias on the sampling error, we conclude that the spatiotemporal sampling errors are rather negligible.

Within an intercomparison study, the OMI TCWV data set proves to be in good agreement to the reference data sets of RSS SSM/I, ERA5, and the ESA WV_cci CDR-2 COMBI in particular over ocean surface. However, over land surface the OMI data set systematically overestimates high TCWV values compared to ERA5 and COMBI by more than 24% especially in the tropical regions affected by frequent cloud cover. Similar results are found from intercomparisons with in situ radiosonde measurements of the IGRA2 data set. The reasons for these overestimations are manifold, but likely due to an overestimation of the OMI TCWV retrieval due to uncertainties in the retrieval input data (surface albedo, cloud information) on the one hand and an underestimation of the reference data due to missing or uncertain observations on the other hand. Nevertheless, the validation also shows that for TCWV $< 25 \, \mathrm{kg} \, \mathrm{m}^{-2}$ good agreement to the reference data can be obtained and also for the case when regions of large uncertainty are filtered. Considering the temporal stability analysis no significant deviation trends could be obtained.

For the cases of ERA5 and RSS SSM/I, the temporal stabilities of less than ±0.1 % per decade meet the "goal" requirements of the latest GCOS report and for the case of COMBI still the "goal" requirement is met. This demonstrates that the OMI TCWV data set is well suited for climate studies.

Altogether, the OMI TCWV data set provides a promising basis for investigations of climate change: on the one hand, it covers a long time series (more than 16 years and with measurements still in operation), and on the other hand, these measurements are based on a single instrument, so that no bias corrections between different sensors need to be taken into account (e.g. in trend analysis studies). Although OMI is affected by degradation effects, we were able to successfully suppress these effects by using Earthshine reference spectra. Furthermore, the data set is based on a retrieval in the visible blue spectral range, where a similar sensitivity for the near-surface layers over ocean and land is given and thus a consistent global data set can be obtained from measurements of only one sensor.

In the future, we plan to complement the data set with TCWV measurements from TROPOMI to ensure the continuation of the data set after the end of the OMI mission. Since the TCWV retrieval can be easily applied to other UV-vis satellite instruments, additional data sets from other instruments from past and present missions such as GOME-1/2 and SCIAMACHY, but

also to future instruments such as Sentinel-5 on MetOp-SG can be created and eventually combined with the OMI TCWV data set taking into account the different instrumental properties (e.g. observation time). This would allow the construction of a data record that extends from 1995 to today. Similarly, a combination of data from low-earth orbit satellites and geostationary satellite instruments such as GEMS, TEMPO or Sentinel-4 could be a promising option to fill temporal gaps in daily observations, but also to investigate (semi-) diurnal cycles of the water vapour distribution.

## 8 Data availability

The MPIC OMI total column water vapour (TCWV) climate data record is available at https://doi.org/10.5281/zenodo.7973889 (Borger et al., 2023).

*Author contributions.* CB performed all calculations for this work and prepared the manuscript together with SB and TW. TW supervised this study.

*Competing interests.* The authors declare that they have no conflict of interest.

*Acknowledgements.* The combined microwave and near-infrared imager based product COMBI was initiated, funded and provided by the Water Vapour project of the ESA Climate Change Initiative, with contributions from Brockmann Consult, Spectral Earth, Deutscher Wetterdienst and the EUMETSAT Satellite Climate Facility on Climate Monitoring (CM SAF). The combined MW and NIR product will be owned by EUMETSAT CM SAF. In particular, we would like to thank Marc Schröder and the ESA CCI WV team for providing the CDR TCWV and common mask data.

## Appendix A: Irradiance based vs. Earthshine SCD

To reduce the across-track biases of the retrieved $H_2O$ SCDs based on a solar reference spectrum, a destriping algorithm can be performed during post-processing. For instance, one way to destripe the swath of an OMI orbit is to

1. calculate the median SCD for each OMI row along-track,

2. calculate the across-track median SCD from the along-track median SCDs,

3. calculate the deviation of the along-track median SCDs from this across-track median SCD,

4. subtract the deviation from the SCDs of the respective OMI row.

For the case of an Earthshine reference this is already implictly accounted for during the spectral analysis, however, one still
has to consider that the Earthshine reference spectrum is not perfectly pristine of the trace gas of interest. For example in our case, although the water vapour concentrations in Antarctica are very low, the Earthshine reference might still be contaminated because of the long light path at such high solar zenith angles.

Figure A1 illustrates the time series of the global monthly mean $H_2O$ SCDs derived from the annual-mean solar irradiance (and destriped following the aforementioned destriping process) and the Earthshine reference for SZA < 80°. Until 2009 the
offset between both SCDs remains constant at values around $0.2 \times 10^{23}$ molec cm$^{-2}$. Between 2009 and 2015 the irradiance based SCDs first decrease and then increase distinctively compared to the Earthshine based SCDs and from 2015 onwards a strong increase in the irradiance based SCDs can be observed. In contrast, the Earthshine SCDs show no jumps or steps and remain at the same magnitude after 2015 and over the complete time range in general.

To get an overview of how the SCD difference (i.e. solar irradiance based minus Earthshine SCD) behaves with time over
the complete OMI swath, Fig. A2 depicts the monthly mean SCD difference for each OMI row. Between 2005 and 2009 the SCD differences remain quite constant for each row, however, after 2009 artefacts arise first at rows 55-60 and then start to expand to other rows and become even stronger. This clearly illustrates that a OMI TCWV product based on a solar irradiance fit cannot be used for trend analyses.

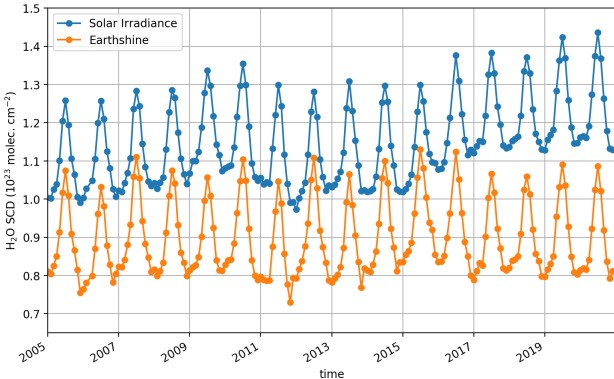

**Figure A1.** Globally averaged monthly-mean of the destriped $H_2O$ SCDs derived from annual-mean solar irradiance and $H_2O$ SCDs derived using the annual Earthshine reference from 2005 until 2020.

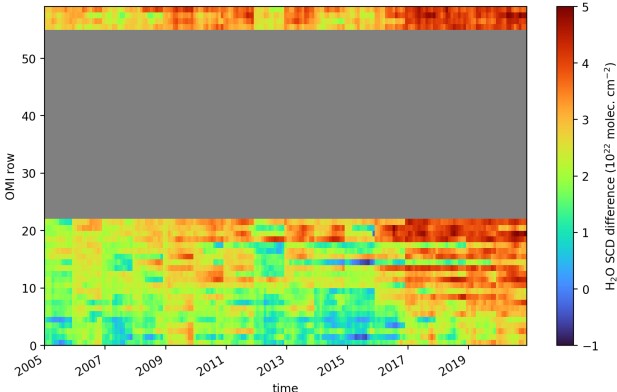

**Figure A2.** Global mean monthly averaged difference between annual-mean irradiance and Earthshine $H_2O$ SCD for each OMI row separately. Only observations with a solar zenith angle $< 80°$ and which are snow- and ice-free are included. Rows affected by the "row-anomaly" (coloured in grey) are excluded for the complete time series.

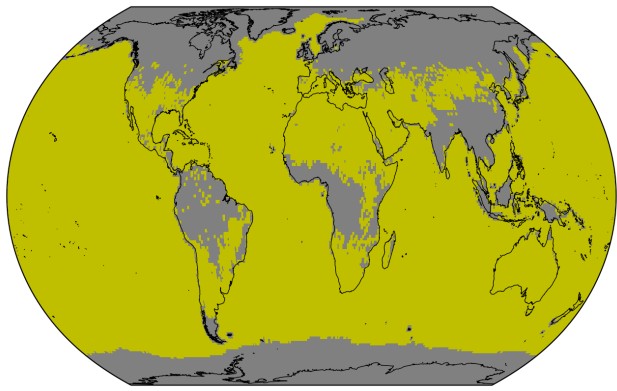

**Figure B1.** "Common mask" of the COMBI data set. Yellow grid cells indicate data points which are accounted for within a temporal stability analysis. Invalid grid cells are coloured grey.

## Appendix B: Intercomparisons taking into account masks and flags

The intercomparison in Sect. 4 also considers regions for which only a small number of measurements are available, for example due to frequent cloud cover or seasonality of the solar zenith angle. On the one hand the small sample size of measurements leads to a higher statistical uncertainty with regard to the monthly mean, and on the other hand also to a non-continuous time series when data are missing for the complete month. Moreover, the errors of the individual measurements are also significantly larger in these regions. With the help of the "common-mask" of the COMBI data set (see Fig. B1), these regions can be

identified and filtered for additional intercomparisons. The "common mask" only considers grid cells for which valid TCWV values were available over all time steps in the COMBI data set in the time period July 2002 to March 2016 (Schröder et al., 2023).

In addition, two flags were created from the MPIC OMI TCWV dataset itself:

1. A static flag for filtering coastlines. For each grid cell, it was checked whether all corners and the centre of the grid cell

are either over ocean or land. If all coordinates are over land, the cell is declared as "land", if all cells are over ocean, it is declared as "ocean" and otherwise it is declared as coastline ("coast"). The resulting map is shown in Fig. B2.

2. A dynamic monthly flag based on the number of measurements to calculate the individual monthly means per grid cell. We have chosen to consider a grid cell as valid if the monthly mean is calculated from more than 100 measurements. This represents a good compromise between global coverage and a good statistic for calculating the monthly mean. Figure B3

shows the fractional coverage for the complete time range of the data set using this mask. Compared to the COMBI mask (see Fig. B1), similar regions are filtered. However, a major advantage of the MPIC mask is that it considers temporal changes, so that the seasonal variability of the atmosphere (e.g. cloud cover and solar zenith angle) is also taken into account when flagging.

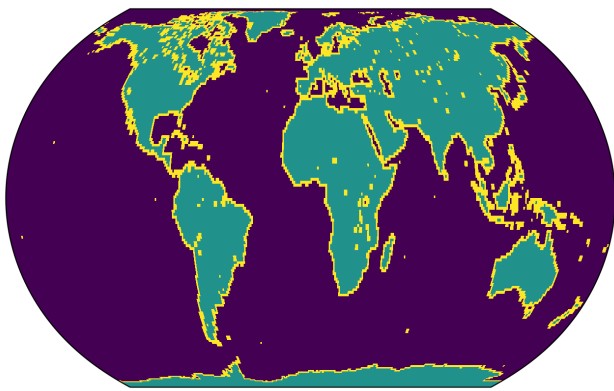

**Figure B2.** Global distribution of the coastline flag of the MPIC OMI TCWV data set.

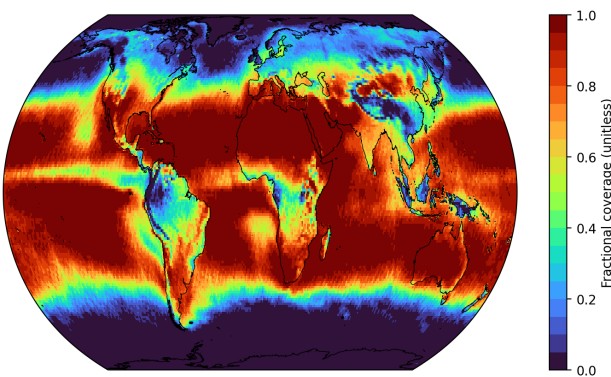

**Figure B3.** Global distribution of the fractional coverage considering the count flag of the MPIC OMI TCWV data set.

The results of the intercomparisons taking into account the COMBI mask and the MPIC OMI flags, respectively, are shown
in Fig. B4 and Fig. B5 for data over ocean and in Fig. B6 and Fig. B7 for data over land. Overall, it can be seen that the mask of
COMBI and the flags of the OMI data set lead to similar changes in the comparisons to the reference data. For all comparisons,
the coefficients of determination for the ODR regression remain at approximately a similar level (i.e. $R^2$ above 0.90) as for the
non-"filtered" comparisons. For the comparisons over ocean hardly any changes are obtained, as the filter is mainly applied
over land surfaces. The differences between the comparisons with the different filters result mainly from the fact that the MPIC
flags filter measurements in the higher latitudes (especially during the winter months).

However, there is a remarkable improvement for the comparison over land: although the fit results of the ODR change only
slightly, the extreme overestimates at high TCWV values are now filtered out and the distributions are now closer to the 1-1
diagonal. Overall, the results for the "filtered" comparison over land also agree very well with the results of the piecewise linear
regression, for which similar slope regression results were found for TCWV < 25 kg m$^{-2}$.

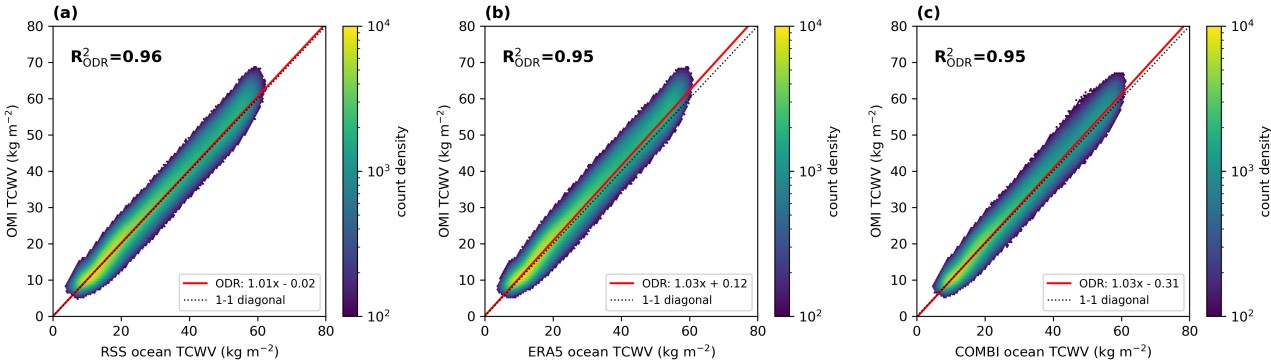

**Figure B4.** Correlation analysis of the OMI TCWV data set and RSS SSM/I, ERA5, and COMBI for data over ocean taking into account only valid grid cells according to "common mask" in Figure B1.

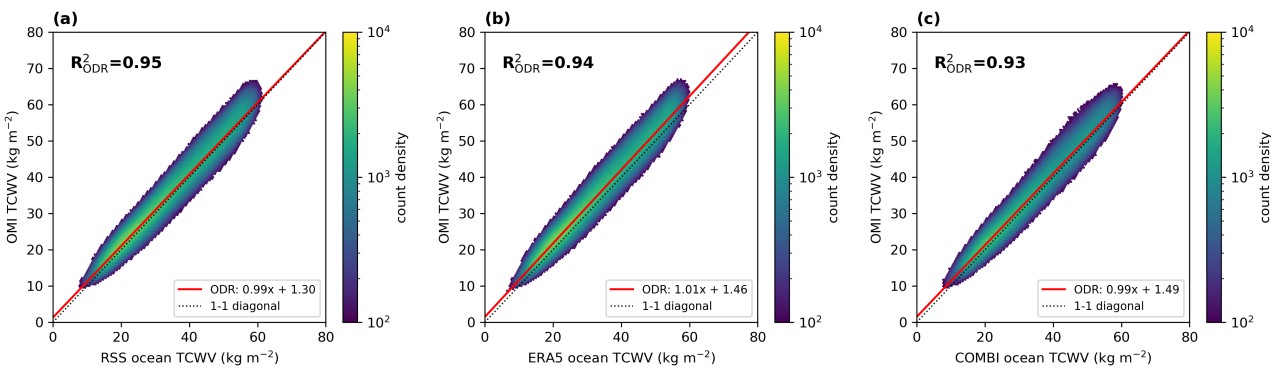

**Figure B5.** Same as Fig. B4, but taking into account only valid grid cells according to coastline and count flag of the MPIC OMI TCWV dataset.

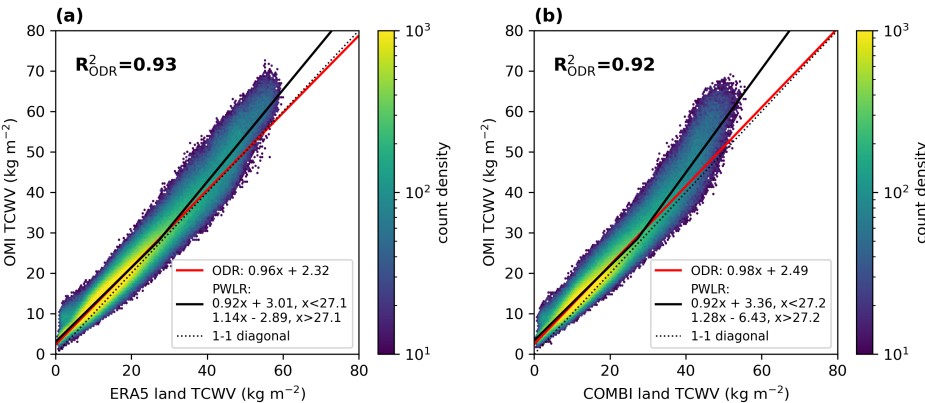

**Figure B6.** Same as Fig. B4, but for data over land. The red solid line represents the ODR results and the solid black line the PWLR results.

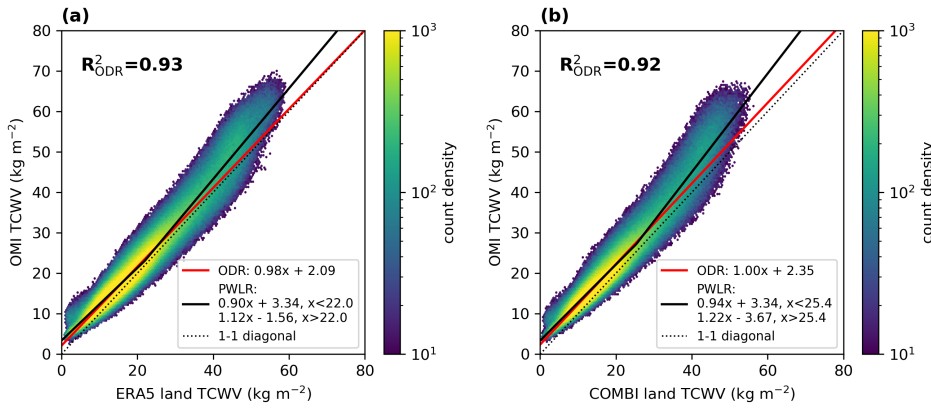

**Figure B7.** Same as Fig. B6, but taking into account only valid grid cells according to coastline and count flag of the MPIC OMI TCWV dataset.

## Appendix C:  Representativeness of row-anomaly filtered data in comparison to full swath

Due to the row anomaly filter, approximately 50% of the complete satellite swath of OMI is not considered in the TCWV data set. This raises the question of how much the monthly mean values would differ if the data of the complete swath were available. To investigate this, we follow the same scheme as in Sect. 3 and use the same ERA5 data as a reference. We select the ERA5 data to match the OMI overpass, once applying the row-anomaly filter and once not. However, in both cases the clear-sky filter based on the OMI cloud information is applied (effective cloud fraction < 20%).

Compared to the clear-sky bias, the deviations are much weaker and no particular spatial patterns are discernible in the global distributions except in the deep Pacific tropics and parts of Southeast Asia (see Fig. C1). Furthermore, the histograms for the absolute and relative deviations in Fig. C2 show a normal distribution for both cases with mean values of $-0.30\,\mathrm{kg\,m^{-2}}$ and -2.1% (and for the median $-0.23\,\mathrm{kg\,m^{-2}}$ and -1.1%). Considering the much larger uncertainties of the OMI TCWV retrievals of typically 20% and more and that the clear-sky bias is almost one order of magnitude larger, the obtained deviations are negligible and thus the monthly means from the RA-filtered data are a good representation compared to the monthly means from the data for a full swath, even though only half of the satellite data is actually used.

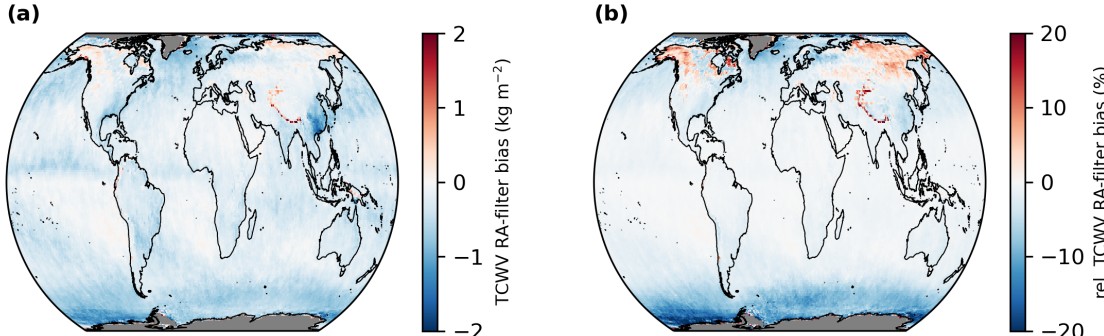

**Figure C1.** Global distributions of the mean differences between row-anomaly (RA) filtered and full swath ERA5 based on the OMI cloud information for the time range January 2005 to December 2020. Panel (a) depicts the absolute differences (i.e. RA-filtered minus full swath) and Panel (b) relative differences (i.e. (RA-filtered minus full swath) / full swath). Grid cells for which no data is available are coloured grey.

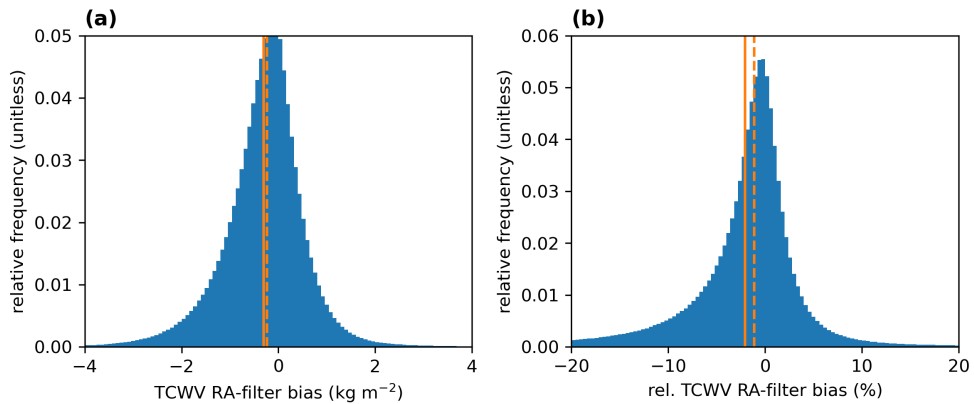

**Figure C2.** Distributions of the absolute differences (RA-filtered minus full swath; Panel a) and relative differences ((RA-filtered minus full swath) / full swath; Panel b) of the monthly mean differences between RA-filtered and full swath ERA5 data based on the OMI cloud information. The solid and dashred orange line indicate the mean and the median of the distributions, respectively.

## Appendix D: Temporal stability analysis with respect to IGRA2

In addition to the global data sets, a stability analysis was also carried out with the IGRA2 radiosonde data. Due to the criterion of temporal coverage, only 62 of the more than 700 IGRA2 stations are left for the analysis. Since almost all of these are located in the northern mid-latitudes, the stability analysis is not globally representative, but the comparison can provide further important independent information.

The course of the temporal stability and the results of the analysis are depicted in Fig. D1. To calculate the temporal stability, a PACF analysis was conducted which revealed that a AR(2) model is most appropriate. Following the same procedure as in Sect. 6, the transformed GLS regression yielded a stability of $+1.33 \pm 1.37\,\%$ per decade. Although this does not fulfill any stability criterion, these results are considerably better than the findings for the COMBI TCWV data set over land (see Fig. 17). Furthermore, it is difficult to determine whether the trend may come from the radiosondes themselves, as it is not clear how regularly the radiosondes are calibrated (e.g. according to the standard of the GCOS Reference Upper Air Network GRUAN (Dirksen et al., 2014)).

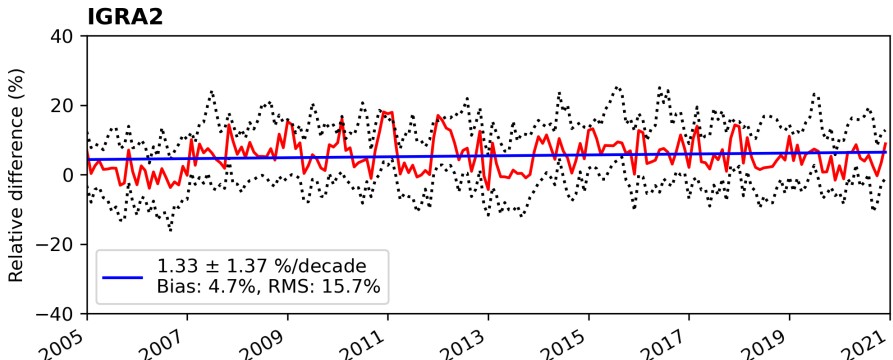

**Figure D1.** Stability analysis of the mean relative deviations of the OMI TCWV data set with respect to IGRA2 radiosonde data for the time range January 2005 to December 2020. Red line: global mean relative deviation; blue line: results of the transformed GLS regression; dotted black line: 25th and 75th percentiles, respectively. The bias and RMS provided in the legends correspond to the time seriues of the global mean deviation for the respective time range.

## Appendix E: Intercomparison to COMBI over full time period

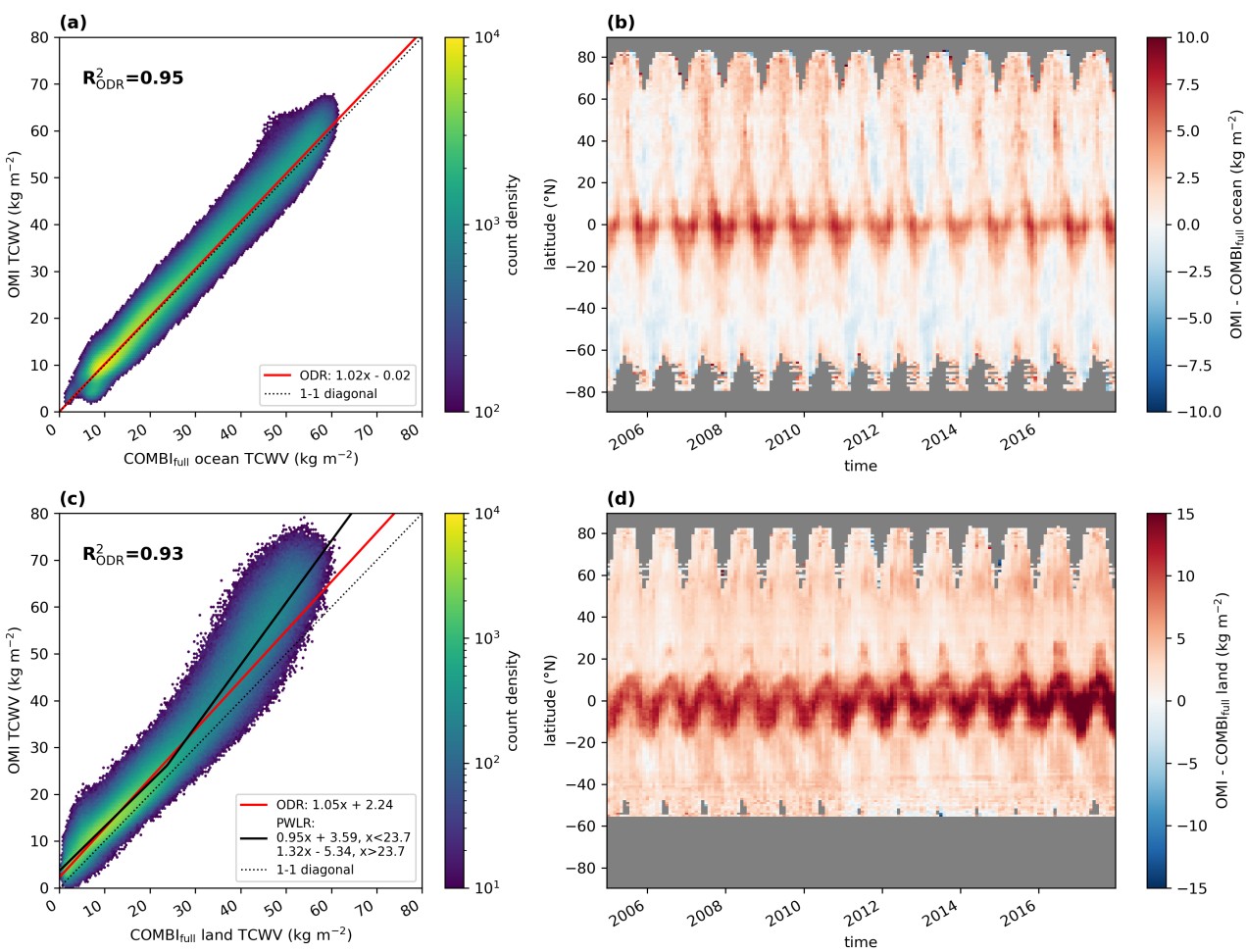

**Figure E1.** Same as Fig. 11, but now with COMBI data for data over ocean (top row) and for data over land (bottom row) for the complete time range.

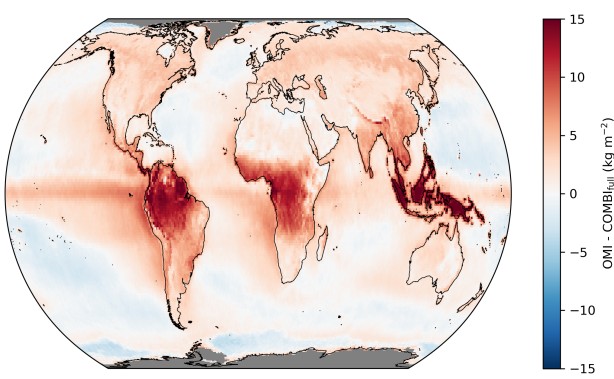

**Figure E2.** Same as Fig.12, but now for COMBI data over the complete time range.

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
