# Peer review of "A 16-year global climate data record of total column water vapour generated from OMI observations in the visible blue spectral range"

_Earth System Science Data, 2021_

## Referee Comment (RC1)

Review of manuscript essd-2021-319 entitled "A 16-year global climate data record of total column water vapour generated from OMI observations in the visible blue spectral range" by Christian Borger, Steffen Beirle, and Thomas Wagner.

General comments

This manuscript presents a total column water vapour (TCWV) data set derived from 16 years of OMI observations. The retrieval method was developed in a previous publication (Borger et al., AMT, 2020). It has been slightly improved in order to meet the long term stability requirements for a climate data record. The manuscript describes briefly the modified aspects of the retrieval algorithm and gives additional details on the data quality control applied for this specific purpose. The latter seems to reject a significant fraction of the raw data, although this number is not indicated. Most of the manuscript is devoted to a comparison/validation analysis of the OMI TCWV results with respect to satellite microwave radiometer data (SSM/I), reanalysis data (ECMWF's ERA5), and the ESA/CCI/CDR-2 water vapour product. The comparison results are fairly documented, including scatter plots, Hovmoeller diagrams and maps, although some synthetic statistics are missing (see the specific comments below). However, the conclusions sound far too optimistic to me, given the poor agreement found between the OMI data and the validation data. Especially, the large positive biases over land and near the coastlines in the tropics are striking and not sufficiently commented or explained. Two main reasons are hypothesized: too low land surface albedo and incorrect cloud information, both leading to an underestimation of the AMF. These paths should be further explored in order to achieved a more reliable product meeting the climate data quality requirements. Although it is shown that the results are improved when a special could mask is used, this is only an artificial way to improve the quality of the product.

Regarding the temporal stability, it is not clear how the significance of the global mean bias, RMSE, and trend differences are established. It seems to me that the numbers are beyond the limits usually required for water vapour climate data (e.g. an error of 0.1%/decade in the global mean TCWV trend represents nearly 20% of the signal). Moreover, the uncertainty due to different time and space sampling with the different reference products should also be quantified.

In conclusion, it is my feeling that the proposed data set has significant defects that are not well understood. I recommend first a more insightful analysis of the error sources, especially over land and, if possible, the elaboration of an improved version of the data set, and second, a more comprehensive discussion of the validation results in a revised version of the manuscript.

Specific comments

More should be said about the "row anomaly" which affects the OMI observations throughout almost the whole period analysed in this paper. Figure A2 shows that a large fraction of "rows" are discarded. Is it sufficient to discard these rows or could adjacent rows also be affected in some way? What is the impact of this screening on the representativeness of the final observations?

Why are two regression methods (OLS and ODR) used? In principle, a single statistic is sufficient, unless the difference of results from the two are discussed, but this is not done in this manuscript. I suggest either to choose one or to better justify the choice of two and analyse the obtained differences.

L77: replace "I and I0" by "I0 and I"

L129: define also Delta_SCD

L144: indicate which fraction of raw data is remaining

L154: "ESA Water Vapour CCI climate data record CDR-2" needs a reference

L154: "For the correlation analysis" is misleading or incorrect if referring to regression analysis. Please reword (e.g. For the intercomparison…)

L155: add a reference for the ODR method

L155-156: "In the case of the ODR it is necessary to use reasonable ratios of the relative errors of the compared data sets instead of using absolute errors in order to obtain meaningful results". This statement needs to be justified by an adequate explanation or reference.

L156-159: these sentences sound in contradiction with the previous statement. Moreover, the sensitivity of the regression results to the relative errors should be discussed in more detail (e.g. in an Appendix) and the choice of 5%, 10%, and 20% for the three dataset (which appear quite arbitrary) should be clearly motivated/discussed.

L169-171: "In general the deviations are quite low with values between +/- 2.5 kg/m2" be more specific in quantifying the differences here, e.g. indicate which fraction of data lie in the range of +/- 2.5 kg/m2, or use quantiles or other statistics (mean, standard deviation, etc.). Note also that the correlation coefficient is not much relevant when the seasonal variations are included.

L173-178: Be more quantitative again, here in the comments on Fig. 4. I would also suggest to include the coastlines of Africa and Indonesia in the list of regions with significant positive deviations.

L176: Be more specific on the impact of the "cold tongue" and "too low albedo" on the observed deviations.

L182: "the slight overestimation of 3-5%": it is not clear what these numbers represent exactly. Is it a mean difference (bias)? Is it computed over all data or only a fraction? (Note that a slope of 1.03 does not mean that all the values are 3% higher, this depends also on the intercept value).

L194: how is the change-point at 26 kg/m2 selected in the piecewise linear regression?

L210: satellite measurements in the thermal infrared are NOT available/reliable in cloudy conditions.

L209-215: I'm not convinced that the ERA5 uncertainty over tropical land areas contributes much to the huge bias observed in the differences (above 10 kg/m2). This idea should be further documented or discarded (also in the Conclusion).

L227: is there any update on the publication of the ESA CDR-2 data set?

L225-254: Similar comments as for ERA5 apply here to the CDR-2 comparison (lack of statistics, etc.).

L261: More details are needed on the linear regression method and significance tests.

L274: The stability requirement for water vapour climate data is rather at the level of 0.3 %/decade (GCOS – 112, April 2007).

Figure 3: the fit results would be more understandable if given as an equation: y = 1.03 x + 0.18 rather than just two numbers.

Figure 3: indicate that the OMI results here are over ocean (it is only obvious if one knows that SSM/I data over only over the oceans).

Figure 4: add the piecewise linear regression lines (mentioned L194) on the plot.

Figure 9: the red dashed lines are not visible in the plots.

---

## Referee Comment (RC2)

**A 16-year global climate data record of total column water vapour generated from OMI observations in the visible blue spectral range**

Christian Borger, Steffen Beirle, and Thomas Wagner

**1. General Comments**

This paper assesses a long term record (2005-2020) of monthly mean total column water vapour (TCWV) from the Ozone Monitoring Instrument (OMI) on board NASAs Aura platform. The authors describe adaptations to an existing algorithm used for ESAs TROPOMI instrument, which is seen as a successor to OMI. This includes the rational for the how and why they switch to using earth shine spectra as the reference in the DOAS retrieval setup. This study goes on to present results from an inter-comparison of TCWV against two addition remote sensing products from RSS and ESA and ERA5 reanalysis.

While this study does discuss issues to do with sampling, I feel this could be expanded especially relating to the clear-sky bias. Work on this has been done within the ESA water vapour CCI project, results from which are relevant to this study and would enhance the discussion around the OMI product performance. Furthermore, there is no mention regarding the quality of the datasets chosen for the inter-comparison exercise. Addition of this information at the beginning of section 3 would help inform a reader unfamiliar with these data sets to why they were used by this study. Finally, what is not clear is to whether this new data record from OMI is meant to complimentary to the existing TROPOMI data set? By this, I mean could the records be used sequentially to bring the time series out the end of the TROPOMI mission? I this is the case, how does the performance of these two record compare?

Overall, I find that this study is of scientific value and recommend it for publication, after all the issues that I have highlighted are addressed.

**2. Specific Comments**

- Section 3: I think the term validation here is incorrect as you are performing inter-comparison of the OMI performance against other gridded products at monthly time scales. For this to be a validation study you would need to perform this on the level 2 swath data against ground truth sites. Alternatively, accurate (fiducial) characterisation of these reference products on monthly time scales would need to be done, and this would be a major undertaking in itself.
- Lines 157-158: What is the assumption you base the relative error estimates on? From the literature, or results not included in this paper? Are you actually describing uncertainties or do you mean errors? Further elaboration here would make this clearer to the reader.
- Section 3.2: For the ERA5 did you take the hourly data and interpolate to the local over pass time **or** the monthly mean data on hourly time steps? Slight rewording to clarify is needed. Additionally, did you consider using the ensemble output which would have given the spread in the reanalysis rather than assigning a relative estimate of the uncertainty?
- Figure 7/B3: The comparison to ESA CCI over land – did you also apply stricter cloud filtering to the OMI data as well as the common mask? The improvement in representativeness can be seen in figure B3 but there could still be additional cloud in the OMI data which is biasing

the data. The common mask from the ESA data will be for 10:00 hrs LST, while with OMI overpasses at 13:30 hrs LST which will have an impact in convective areas. Finally, is there an improvement in the Hovmöller time series when the common mas is applied?

**3. Technical Comments**

- Line 38: the reference Susskind et al. 2003 is for joint microwave and infrared retrievals from AIRS. Therefore, is not an explicit reference for IR water vapour retrievals. There is also an extra ')' on line 39 after the reference, did you mean to have the 2003 in-cased in parenthesises?
- Line 39: both your references here are for near infrared retrievals from MERIS, missing a shortwave infrared reference e.g. SCIAMACHY (2.3 µm), GOSAT (1.6 +2.1 µm), or TROPOMI (2.3 µm).

---

## Author Comment (AC1)

We would like to thank the referee for reviewing our manuscript. Below we reply to the issues raised by the referee, where

blue repeats the reviewer's comments,

black is used for our reply,

*and green italics is used for modified text and new text added to the manuscript.*

Review of manuscript essd-2021-319 entitled "A 16-year global climate data record of total column water vapour generated from OMI observations in the visible blue spectral range" by Christian Borger, Steffen Beirle, and Thomas Wagner.

General comments

This manuscript presents a total column water vapour (TCWV) data set derived from 16 years of OMI observations. The retrieval method was developed in a previous publication (Borger et al., AMT, 2020). It has been slightly improved in order to meet the long term stability requirements for a climate data record. The manuscript describes briefly the modified aspects of the retrieval algorithm and gives additional details on the data quality control applied for this specific purpose. The latter seems to reject a significant fraction of the raw data, although this number is not indicated. Most of the manuscript is devoted to a comparison/validation analysis of the OMI TCWV results with respect to satellite microwave radiometer data (SSM/I), reanalysis data (ECMWF's ERA5), and the ESA/CCI/CDR-2 water vapour product. The comparison results are fairly documented, including scatter plots, Hovmoeller diagrams and maps, although some synthetic statistics are missing (see the specific comments below). However, the conclusions sound far too optimistic to me, given the poor agreement found between the OMI data and the validation data. Especially, the large positive biases over land and near the coastlines in the tropics are striking and not sufficiently commented or explained. Two main reasons are hypothesized: too low land surface albedo and incorrect cloud information, both leading to an underestimation of the AMF. These paths should be further explored in order to achieved a more reliable product meeting the climate data quality requirements. Although it is shown that the results are improved when a special could mask is used, this is only an artificial way to improve the quality of the product.

Regarding the temporal stability, it is not clear how the significance of the global mean bias, RMSE, and trend differences are established. It seems to me that the numbers are beyond the limits usually required for water vapour climate data (e.g. an error of 0.1%/decade in the global mean TCWV trend represents nearly 20% of the signal). Moreover, the uncertainty due to different time and space sampling with the different reference products should also be quantified.

Many thanks for pointing out the GCOS requirements! For our value of 1.0%/decade we have followed the User Requirements of the ESA CCI WV (https://climate.esa.int/media/documents/Water_Vapour_cci_D1.1_URD_v3.0.pdf). If we understand correctly, the value of 0.3%/decade in the GCOS document refers to radiosonde measurements or their WVMR measurements (see requirement tables in Appendix 1 of the GCOS document suggested below in the Specific Comments). Please also note that the requirement mentioned refers to the global mean. However, we are also interested in regional trends, which usually have significantly higher magnitudes.

In addition, we determined the sampling error (and the clear-sky bias) and added the following text to the revised version:

[revised manuscript text omitted]

Moreover, we also investigated trends in the clear-sky bias (which has the largest impact on the sampling error) and obtained absolute trends between +-0.04 kg/m^2 per year (and -0.002kg/m^2 per year on global average), which is one order of magnitude smaller than typical TCWV trends (see e.g. Borger et al., 2022).

In conclusion, it is my feeling that the proposed data set has significant defects that are not well understood. I recommend first a more insightful analysis of the error sources, especially over land and, if possible, the elaboration of an improved version of the data set, and second, a more comprehensive discussion of the validation results in a revised version of the manuscript.

Specific comments

More should be said about the "row anomaly" which affects the OMI observations throughout almost the whole period analysed in this paper. Figure A2 shows that a large fraction of "rows" are discarded. Is it sufficient to discard these rows or could adjacent rows also be affected in some way? What is the impact of this screening on the representativeness of the final observations?
The row anomaly is a dynamic artefact and initially spread from a few isolated rows over a large area of the detector, affecting about 50% of the swath. However, it is observed that it seems to have stabilised or not changed much for a few years (see e.g. Figure 22 in

Schenkeveld et al., 2017). Based on the daily monitoring of the instrument and the rigid row-anomaly screening, we can therefore at least assume that we are filtering the very largest part of the row anomaly to the best of our knowledge, although we cannot say one hundred percent that other rows are slightly affected.

We investigated the extent to which the monthly means would change if the full swath had been taken into account and concluded that the impact is almost an order of magnitude smaller than other uncertainties (e.g. clear-sky bias). The following section is added to the appendix:

*Due to the row anomaly filter, approximately 50% of the complete satellite swath of OMI is not considered in the TCWV data set. This raises the question of how much the monthly mean values would differ if the data of the complete swath were available. To investigate this, we follow the same scheme as in Sect. 3 and use the same ERA5 data as a reference. We select the ERA5 data to match the OMI overpass, once applying the row-anomaly filter and once not. However, in both cases the clear-sky filter based on the OMI cloud information is applied (effective cloud fraction < 20%).*

*Compared to the clear-sky bias, the deviations are much weaker and no particular spatial patterns are discernible in the global distributions except in the deep Pacific tropics and parts of Southeast Asia (see Fig. C1). Furthermore, the histograms for the absolute and relative deviations in Fig. C2 show a normal distribution for both cases with mean values of -0.30 kg m-2 and -2.1% (and for the median -0.23 kg m-2 and -1.1%). Considering the much larger uncertainties of the OMI TCWV retrievals of typically 20% and more and that the clear-sky bias is almost one order of magnitude larger, the obtained deviations are negligible and thus the monthly means from the RA-filtered data are a good representation compared to the monthly means from the data for a full swath, even though only half of the satellite data is actually used.*

[Figure]

*Figure C1. Global distributions of the monthly mean differences between row-anomaly (RA) filtered and full swath ERA5 based on the OMI cloud information for the time range January 2005 to December 2020. Panel (a) depicts absolute differences (i.e. RA-filtered minus full swath) and Panel (b) relative differences (i.e. (RA-filtered minus full swath) / full swath). Grid cells for which no data is available are coloured grey.*

[Figure]

*Figure C2. Distributions of the absolute differences (RA-filtered minus full swath; Panel a) and relative differences ((RA-filtered minus full swath) / full swath; Panel b) of the monthly mean differences between RA-filtered and full swath ERA5 data based on the OMI cloud information. The solid and dashed orange line indicate the mean and the median of the distributions, respectively.*

So, although about 50% of the orbit is missing, this still covers a swath of about 1300km and is thus still larger than the swaths of GOME-1, SCIAMACHY or GOME-2A (all around 960km) or in the order of magnitude of SSMI (about 1394km). Thus, OMI still achieves a complete coverage of the Earth about every 2-3 days, which should provide enough observational data for a good representativeness in the case of a monthly mean (see also the good agreement with the reference data).
We added this information to the revised manuscript:
*So while about 50% of the orbit is missing because of the RA-filter, the remaining data still cover an "effective" swath of about 1300 km and is thus still larger than the swaths of GOME-1, SCIAMACHY, or GOME-2A (all about 1300 km) or of the order of SSM/I (about 1394 km). Thus, OMI still achieves complete coverage of the Earth about every 2-3 days, which should provide enough observational data for good representativeness in case of a monthly mean (see also Appendix C and the good agreement to the reference data in Sect. 4).*

Why are two regression methods (OLS and ODR) used? In principle, a single statistic is sufficient, unless the difference of results from the two are discussed, but this is not done in this manuscript. I suggest either to choose one or to better justify the choice of two and analyse the obtained differences.
We decided to use only the ODR, but also to show the results of the PWLF regression in the scatterplots instead.

L77: replace "I and I0" by "I0 and I"
We replaced the terms accordingly.

L129: define also Delta_SCD
We have added that Delta_SCD is the offset between Earthshine and normal SCD.

L144: indicate which fraction of raw data is remaining
If we only take the data that is already filtered according to the row anomaly as a basis, approx. 30% remains. If we take all the data of an orbit as a basis, approx. 12% remain.

However, this also includes pixels above the polar regions for which a spectral analysis is not possible or does not make sense due to the high noise.
We added the following text:
*In total, this leaves about 30% of data from an RA-filtered orbit and about 12% of data from a complete orbit.*

L154: "ESA Water Vapour CCI climate data record CDR-2" needs a reference
At the moment, no reference for the data set is available yet.

L154: "For the correlation analysis" is misleading or incorrect if referring to regression analysis. Please reword (e.g. For the intercomparison...)
We have changed the phrase accordingly.

L155: add a reference for the ODR method
We have added Cantrell (2008) as a reference.

L155-156: "In the case of the ODR it is necessary to use reasonable ratios of the relative errors of the compared data sets instead of using absolute errors in order to obtain meaningful results". This statement needs to be justified by an adequate explanation or reference.
Based on the descriptions of Cantrell (2008), one sees in equation (5) in his paper that the slope depends on a parameter W_i, which relates the uncertainties w_x and w_y of x and y to each other (see formula below).

$$b = \bar{y} - m\,\bar{x} \quad m = \frac{\sum W_i \beta_i V_i}{\sum W_i \beta_i U_i}$$

$$\bar{x} = \sum W_i x_i \Big/ \sum W_i \quad \bar{y} = \sum W_i y_i \Big/ \sum W_i$$

$$U_i = x_i - \bar{x} \quad V_i = y_i - \bar{y} \quad W_i = \frac{w_{xi} w_{yi}}{w_{xi} + m^2 w_{yi} - 2 m r_i \alpha_i}$$  \quad (5)

$$\beta_i = W_i \left[ \frac{U_i}{w_{yi}} + \frac{m V_i}{w_{xi}} - (m U_i + V_i)\frac{r_i}{\alpha_i} \right] \quad \alpha_i = \sqrt{w_{xi} w_{yi}}$$

In the case that the error in y is significantly larger than in x, the ODR approaches ordinary linear regression.

Cantrell, C. A.: Technical Note: Review of methods for linear least-squares fitting of data and application to atmospheric chemistry problems, Atmos. Chem. Phys., 8, 5477–5487, https://doi.org/10.5194/acp-8-5477-2008, 2008.

L156-159: these sentences sound in contradiction with the previous statement. Moreover, the sensitivity of the regression results to the relative errors should be discussed in more detail (e.g. in an Appendix) and the choice of 5%, 10%, and 20% for the three dataset (which appear quite arbitrary) should be clearly motivated/discussed.
To motivate our choice, we have added the following text to the revised version:
*Mears et al. (2015) found that the uncertainty of daily microwave TCWV observations for TCWV=10 kg m-2 was around 1 kg m-2 and for TCWV = 60 kg m-2 around 2-4 kg m-2. Hence, we assume that the uncertainty of the RSS data set is 5% or at least 1 kgm−2. For*

*ERA5 and ESA CDR-2 we can assume similar uncertainties over ocean, since the TCWV values there are also mainly based on microwave observations. Unfortunately, no uncertainties are provided for TCWV over land. Thus, for the sake of simplicity, we assume that the relative errors of the reference data sets over land are twice as high as over ocean, i.e. 10% or at least 2 kg m-2. For the OMI TCWV data set we assume an uncertainty of 20% (Borger et al., 2020), but at least 2kg m-2. We also tested other error assumptions and it turned out that the exact choice of errors is negligible for the regression results as long as the ratio of uncertainties remains similar.*

L169-171: "In general the deviations are quite low with values between +/- 2.5 kg/m2" be more specific in quantifying the differences here, e.g. indicate which fraction of data lie in the range of +/-2.5 kg/m2, or use quantiles or other statistics (mean, standard deviation, etc.). Note also that the correlation coefficient is not much relevant when the seasonal variations are included.

We added the information of the mean bias together with the standard deviation for the comparison to every data set and also provide this information for the tropics (-20°N – 20°N) and for the extratropics.

With regard to the correlation coefficient, we cannot fully agree, as it includes spatial variation in addition to temporal variation: namely, if we reverse the latitudes, we only obtain a correlation of R=0.63 for RSS and R=0.45 for ERA5 over land.

L173-178: Be more quantitative again, here in the comments on Fig. 4. I would also suggest to include the coastlines of Africa and Indonesia in the list of regions with significant positive deviations.

As mentioned above, we now provide the mean bias and the standard deviation. Moreover, we rephrased the sentence:

*Consistent with the findings from Fig. 7 highest positive deviations can be found in the tropical Pacific ocean and near the coastlines of South America, Africa, and Indonesia whereas […]*

L176: Be more specific on the impact of the "cold tongue" and "too low albedo" on the observed deviations.

The area of the "cold tongue" is often affected by low maritime clouds (cloud top height at approx. 1km). Since the highest water vapour concentration are found in the lower troposphere or boundary layer, deviations in the AMF of the order of 10% can occur even with slightly deviating cloud heights of a few 100m.

In the area of Central America and the west coast of Africa, the albedo is influenced by the absorption by phytoplankton (Kleipool et al., 2008), which may not have been optimally corrected during the creation of the LER or ensures that already low albedo values can lead to further small deviations, which are then again large in relative terms (e.g. with albedo values of 0.05 to 0.04).

We rephrased the text as follows:

*In the case of the tropical Pacific ocean the distribution of the systematic positive deviations matches quite well regions of cold water or of the so called "cold tongue" which is frequently affected by low clouds. Since the highest water vapour concentrations occur in the lower troposphere, small deviations of a few 100m in cloud height can have relatively large effects on the AMF. In the case of Central America or Atlantic ocean, a too low albedo due to additional absorption by phytoplankton (Kleipool et al., 2008) could explain the systematic positive deviations.*

L182: "the slight overestimation of 3-5%": it is not clear what these numbers represent exactly. Is it a mean difference (bias)? Is it computed over all data or only a fraction? (Note that a slope of 1.03 does not mean that all the values are 3% higher, this depends also on the intercept value).

Many thanks for this hint! Indeed, we have not expressed our approach clearly enough. By 3-5% overestimation, we are referring to the slope of the fit line. Regarding the y-axis intercept, we will explicitly mention it if it is larger than the minimum assumed uncertainty (1kg over ocean, 2kg over land). For the ocean comparisons, the offsets are less than +-0.25kg/m^2, so they are negligible and thus the slope is sufficient as the sole indicator of over- or underestimation. For the comparisons for the data over land, however, they are systematically higher than the minimum uncertainty, so we have revised the text of the respective comparisons:

*For data over land, the picture is different: although the ODR gives similar results for the slope as for data over ocean, the distribution in the 2D histogram (Fig. 9c) shows particularly strong positive deviations of approximately +10 kg m−2 at high TCWV values and an overall systematic offset of around +1.43 kg m−2.Within the PWLF analysis we find a good agreement to the reference data for TCWV values up to about 25 kg m−2 (which represents approximately 74% of all data points) with slopes of around 0.96. However, for higher TCWV values we find distinctive positive overestimations of up to 24%. Nevertheless, even for low TCWV values a systematic offset of approximately +2.52 kg m−2 is obtained. [...]*

*Similar to the intercomparison of ERA5, the intercomparison over land (Fig. 11c) shows roughly similar ODR fit results as over ocean, but here we also find striking positive deviations for high TCWV values and an overall positive offset of 2.41 kg m−2. Again, when applying a piecewise linear regression analysis we obtain good agreement with slopes of around 0.95 for TCWV values to about 25 kgm−2 but still a distinctive positive offset of 3.73 kg m−2 for low TCWV values and distinctive overestimations of up to 33% for higher TCWV values, which is even higher than for the comparison to ERA5.*

L194: how is the change-point at 26 kg/m2 selected in the piecewise linear regression?

The change point is automatically determined by a non-linear least-squares fit.

L210: satellite measurements in the thermal infrared are NOT available/reliable in cloudy conditions.

We have reworded the sentence as follows:

*[...] satellite measurements (or none at all in the thermal infrared) [...]*

L209-215: I'm not convinced that the ERA5 uncertainty over tropical land areas contributes much to the huge bias observed in the differences (above 10 kg/m2). This idea should be further documented or discarded (also in the Conclusion).

The regions in question are highly affected by quasi-permanent cloud cover, so observations are systematically missing and there may be a clear-sky bias, which can be in the order of a few kg/m^2 (see also Sect. 3 in the revised manuscript). And even if radiances are assimilated into cloudy-sky scenarios, their uncertainty is still large, as the radiative transfer of cloudy pixels is highly complex (e.g. Li et al., 2016). Especially even in the ESA WV_cci CDR these regions are flagged, although MODIS should have enough observations available for good statistics. We conclude that the large deviations in the tropics cannot, of course, be completely attributed to the uncertainties in ERA5, but they are not so small as to be negligible either.

Li, J., Wang, P., Han, H. et al. On the assimilation of satellite sounder data in cloudy skies in numerical weather prediction models. J Meteorol Res 30, 169–182 (2016). https://doi.org/10.1007/s13351-016-5114-2

L227: is there any update on the publication of the ESA CDR-2 data set?
To the best of our knowledge, no publication is available at the moment.

L225-254: Similar comments as for ERA5 apply here to the CDR-2 comparison (lack of statistics, etc.).
See comment above about added statistics.

L261: More details are needed on the linear regression method and significance tests.
For the analysis, we use an ordinary least-squares fit, with the significance test or p-value based on a two-sided Students t-test (see also https://docs.scipy.org/doc/scipy/reference/generated/scipy.stats.linregress.html). We added this information in the revised manuscript as follows:
*[…] and then calculate temporal trends of these deviations using linear ordinary linear least-squares regression following the approach of Danielczok and Schröder (2017) and Beirle et al. (2018) and assess the significance of the results based on a two-sided Student's t-test.*

L274: The stability requirement for water vapour climate data is rather at the level of 0.3 %/decade (GCOS – 112, April 2007).
See comment above in the General Comments section.

Figure 3: the fit results would be more understandable if given as an equation: y = 1.03 x + 0.18 rather than just two numbers.
We have changed the legends in the figures accordingly.

Figure 3: indicate that the OMI results here are over ocean (it is only obvious if one knows that SSM/I data over only over the oceans).
We added in the Figure caption that the results correspond to data over ocean.

Figure 4: add the piecewise linear regression lines (mentioned L194) on the plot.
We have added information of the PWLF regression results in all relevant figures.

Figure 9: the red dashed lines are not visible in the plots.
We revised Figure 9 and removed the dashed red lines.

---

## Author Comment (AC2)

We would like to thank the referee for reviewing our manuscript and for the many useful comments and suggestions. Below we reply to the issues raised by the referee, where
blue repeats the reviewer's comments,
black is used for our reply,
*and green italics is used for modified text and new text added to the manuscript.*

A 16-year global climate data record of total column water vapour generated from OMI observations in the visible blue spectral range
Christian Borger, Steffen Beirle, and Thomas Wagner

1. General Comments
This paper assesses a long term record (2005-2020) of monthly mean total column water vapour (TCWV) from the Ozone Monitoring Instrument (OMI) on board NASAs Aura platform. The authors describe adaptations to an existing algorithm used for ESAs TROPOMI instrument, which is seen as a successor to OMI. This includes the rational for the how and why they switch to using earth shine spectra as the reference in the DOAS retrieval setup. This study goes on to present results from an inter-comparison of TCWV against two addition remote sensing products from RSS and ESA and ERA5 reanalysis.

While this study does discuss issues to do with sampling, I feel this could be expanded especially relating to the clear-sky bias. Work on this has been done within the ESA water vapour CCI project, results from which are relevant to this study and would enhance the discussion around the OMI product performance.

Many thanks for this important comment! We agree with the reviewer and have decided to estimate the clear-sky bias within our data set. Moreover, following the suggestions from Reviewer #1, we also estimate sampling errors. As such, we have added the following text to the revised version:

[revised manuscript text omitted]

*Furthermore, there is no mention regarding the quality of the datasets chosen for the inter-comparison exercise. Addition of this information at the beginning of section 3 would help inform a reader unfamiliar with these data sets to why they were used by this study.*

We thank the reviewer for this suggestion and have restructured Section 3 accordingly. Moreover, we added the following text to the beginning of Sect. 3:

*To evaluate the overall quality of the OMI TCWV data set, we conducted an intercomparison study for which we use the merged, 1-degree total precipitable water (TPW) data set version 7 from Remote Sensing Systems (RSS) (Mears et al., 2015; Wentz, 2015), TCWV data from the reanalysis model ERA5 (Hersbach et al., 2019, 2020), and the ESA Water_Vapour_CCI (WV_cci) climate data record CDR-2 as reference.*

*The RSS data set consists of merged geophysical ocean products whereby the values are retrieved from various passive satellite microwave radiometers. These microwave radiometers have been intercalibrated at the brightness temperature level and the ocean products have been produced using a consistent processing methodology for all sensors (more details in Wentz, 2015; Mears et al., 2015). The major advantages of microwave TCVW retrievals are their high precision and accuracy and that they are insensitive to clouds, so that TCWV values can also be retrieved even under cloudy-sky conditions. A disadvantage, however, is that these retrievals are (mostly) only available over the ocean surface.*

*Thus, we also compare the OMI TCWV data to the ESAWV_cci CDR-2. At the moment of preparation of this manuscript, the CDR-2 is a beta-version of the combined microwave and near-infrared imager based TCWV data record (COMBI). The CDR combines microwave and near-infrared imager based TCWV over the ice-free ocean as well as over land, coastal ocean and sea-ice, respectively. The data record relies on microwave observations from SSM/I, SSMIS, AMSR-E and TMI, partly based on a fundamental climate data record (Fennig et al., 2020) and on near-infrared observations from MERIS, MODIS-Terra and OLCI (Danne et al., 2022).*

*Within comparisons between different satellite data sets a major drawback is the influence of sampling errors due to different observation times, pixel footprint sizes or orbit patterns. To minimise this source of error, data from reanalysis models are useful. ERA5 is the fifth generation ECMWF reanalysis (Hersbach et al., 2020) and combines model data with in situ and remote sensing observations from various different measurement platforms. For our purpose, we use the "monthly averaged reanalysis by hour of day" from the Copernicus Climate Data Store on a 1°x 1° grid. To account for OMI's observation time (around 13:30 LT), we first calculate the local time for each longitude in the ERA5 data set, then select the TCWV data for the time period between 13:00-14:00 LT and finally merge the selected data. […]*

*Finally, what is not clear is to whether this new data record from OMI is meant to complimentary to the existing TROPOMI data set? By this, I mean could the records be used sequentially to bring the time series out the end of the TROPOMI mission? I this is the case, how does the performance of these two record compare?*

Indeed, in the future we plan to merge the TCWV datasets of OMI and TROPOMI or to continue the OMI dataset using TROPOMI data. For the time being, however, we are refraining from doing so:

- the TROPOMI cloud algorithms are not yet fully developed and there are currently repeated jumps in the TROPOMI TCWV dataset (see for example Küchler et al., 2021).

- the OMI radiances have been processed with the TROPOMI processor since 2022 and will also be reprocessed with it after the end of the mission. This should lead to an improvement of the irradiance and radiance spectra, so that it may be possible to switch from an Earthshine fit to a solar irradiance fit.
- OMI will soon run out of fuel and thus the mission will end soon (2023 or 2024). By then, there should also be enough overlap between OMI and TROPOMI.

Overall, I find that this study is of scientific value and recommend it for publication, after all the issues that I have highlighted are addressed.

2. Specific Comments
- Section 3: I think the term validation here is incorrect as you are performing inter-comparison of the OMI performance against other gridded products at monthly time scales. For this to be a validation study you would need to perform this on the level 2 swath data against ground truth sites. Alternatively, accurate (fiducial) characterisation of these reference products on monthly time scales would need to be done, and this would be a major undertaking in itself.

We agree that the term "validation" is not adequate and will instead refer to an "intercomparison study".

- Lines 157-158: What is the assumption you base the relative error estimates on? From the literature, or results not included in this paper? Are you actually describing uncertainties or do you mean errors? Further elaboration here would make this clearer to the reader.

For SSMI, we followed the results of Mears et al. (2015), who found that the uncertainty for TCWV = 10mm was around 1mm and for TCWV = 60mm around 2-4mm. Thus, we have revised the uncertainties again and specify that the uncertainty is 5% or at least 1mm.

For ERA5 and ESA CDR we can assume similar uncertainties over ocean, since the TCWV values there are also mainly based on microwave observations. For the ESA CDR, "average retrieval uncertainties" are given, but these are unrealistically low over land (<0.1kg/m^2). Therefore, we decided to make a compromise and, for simplicity's sake, set the uncertainty about twice as high as the uncertainty over ocean.

We added the following text to the revised manuscript:

*Mears et al. (2015) found that the uncertainty daily microwave TCWV observations for TCWV=10 kg m-2 was around 1 kg m-2 and for TCWV = 60 kg m-2 around 2-4 kg m-2. Hence, we assume that the uncertainty of the RSS data set is 5% or at least 1 kgm−2. For ERA5 and ESA CDR-2 we can assume similar uncertainties over ocean, since the TCWV values there are also mainly based on microwave observations. Unfortunately, no uncertainties are provided for TCWV over land. Thus, for the sake of simplicity, we assume that the relative errors of the reference data sets over land are twice as high as over ocean, i.e. 10% or at least 2 kg m-2. For the OMI TCWV data set we assume an uncertainty of 20% (Borger et al., 2020). We also tested other variants of error assumptions and it turned out that the exact choice of errors is negligible for the regression results as long as the ratio of uncertainties remains similar.*

- Section 3.2: For the ERA5 did you take the hourly data and interpolate to the local over pass time or the monthly mean data on hourly time steps? Slight rewording to clarify is needed. Additionally, did you consider using the ensemble output which would have given the spread in the reanalysis rather than assigning a relative estimate of the uncertainty?

For the ERA5 data we used the "Monthly averaged reanalysis by hour of day" from the Copernicus climate data store (CDS). Starting from the OMI overpass time (13:30LT), the local time was determined for each longitude and then the TCWV data for the period 13:00 to 14:00LT were selected and merged.

We added the following text to the revised manuscript:

*For our purpose, we use the "monthly averaged reanalysis by hour of day" from the Copernicus Climate Data Store on a 1° x 1° grid. To account for OMI's observation time (around 13:30 LT), we first calculate the local time for each longitude in the ERA5 data set, then select the TCWV data for the time period between 13:00-14:00 LT and finally merge the selected data.*

Regarding the use of ERA5 ensemble data, we have to admit that we are not experts in this field and therefore cannot completely understand how the different ensemble members come about. Nevertheless, we have taken a look at the ensemble data and calculated the ensemble spread and the relative spread as follows:

Spread = max(ensemble) - min(ensemble)

Relative spread = Spread / mean(ensemble)

[Figure]

Fig.RC1: Spread (left) and relative spread (right panel) of ERA5 TCWV ensemble data at around 13:30LT derived from 3-hourly resolved monthly mean by hour of day TCWV data for the time range January 2005 to December 2020. The local time was determined by longitude. Then the data was selected for a time between 12:00 and 15:00LT and finally merged.

In our opinion, however, these spreads underestimate the actual uncertainty of ERA5: Over the ocean, the uncertainties are smaller than in the SSM/I (see comment above), although similar input data should be used. And over land they are in some places only slightly larger than over the ocean, which is not entirely understandable, since much less observational data is available than over the ocean, or over land the (satellite) measurements typically have a larger uncertainty.

Therefore, we believe that our assumption for the uncertainty, is a good estimate of the uncertainties in ERA5.

- Figure 7/B3: The comparison to ESA CCI over land – did you also apply stricter cloud filtering to the OMI data as well as the common mask? The improvement in representativeness can be seen in figure B3 but there could still be additional cloud in the OMI data which is biasing the data. The common mask from the ESA data will be for 10:00 hrs LST, while with OMI overpasses at 13:30 hrs LST which will have an impact in convective areas. Finally, is there an improvement in the Hovmöller time series when the common mas is applied?

Following the suggestion of the reviewer, we have also applied a more stringent cloud filter (CF<5% instead of CF<20%) and have carried out the comparisons with it. For the sake of simplicity, we only show the comparisons with ERA5.

When looking at the difference maps for each season, a clear reduction of the systematic overestimation in the tropics or in the Amazon region can be seen on average, but we now also see clear underestimation in areas with frequent cloud cover (e.g. India or Southeast Asia). This indicates that such a stringent filter leads to a strong clear-sky bias. Furthermore, the filter leads to many gaps in the TCWV data set, so that it would no longer be optimal for time series studies.

Therefore, we think that the current configuration with CF<20% is a good compromise between spatiotemporal coverage and quality.

[Figure]

Fig. RC2: Global distributions of the absolute differences between the OMI TCWV data with a stricter cloud filter (eCF < 5%) and ERA5 for winter (DJF), spring (MAM), summer (JJA), and autumn (SON). Grid cells for which no data is available are coloured grey.

[Figure]

Fig. RC3: Intercomparison between monthly mean TCWV from OMI (with the stricter cloud filter) and ERA5 for data over ocean (top row) and land (bottom row). Panel (a) and (c) illustrate a 2D histogram in which the colour indicates the count density; the red solid line represents the results of the orthogonal distance regression (ODR) and the solid black line the results of the piecewise linear regression (PWLF). The results of the respective fits are given in the bottom right box and the correlation coefficient in the top left corner. The dashed black line indicates the 1-to-1 diagonal. Panels (b) and (d) depict the TCWV difference of OMI minus ERA5 within the latitude-time space; reddish colours indicate an overestimation, blueish colours an underestimation of the OMI TCWV data set.

**3. Technical Comments**

- Line 38: the reference Susskind et al. 2003 is for joint microwave and infrared retrievals from AIRS. Therefore, is not an explicit reference for IR water vapour retrievals. There is also an extra ')' on line 39 after the reference, did you mean to have the 2003 in-cased in parenthesises?

Thank you for the clarification regarding the Susskind et al. reference! We have added the references of Schlüssel et al. (2005) and Schneider and Hase (2011) as examples of TIR retrievals.

- Line 39: both your references here are for near infrared retrievals from MERIS, missing a shortwave infrared reference e.g. SCIAMACHY (2.3 μm), GOSAT (1.6 +2.1 μm), or TROPOMI (2.3 μm).

We have added the references of Schrijver et al. (2009), Dupuy et al. (2014) and Schneider et al. (2020) as examples of SWIR retrievals.

---

## Referee Report (RR1)

2nd review of manuscript "essd-2021-319-manuscript-version4.pdf" entitled "A 16-year global climate data record of total column water vapour generated from OMI observations in the visible blue spectral range" by Christian Borger, Steffen Beirle, and Thomas Wagner.

General comments

I acknowledge for the additional work done by the authors to address the questions of time and space sampling and clear-sky bias. However, I cannot find the answers to my main criticism and recommendations, which are still pending. Let me recall them:

"the conclusions sound far too optimistic to me, given the poor agreement found between the OMI data and the validation data. Especially, the large positive biases over land and near the coastlines in the tropics are striking and not sufficiently commented or explained."

"I recommend first a more insightful analysis of the error sources, especially over land and, if possible, the elaboration of an improved version of the data set, and second, a more comprehensive discussion of the validation results in a revised version of the manuscript.

The biases in the proposed data set are striking, see the reddish patterns in Figs. 3,4,5,6,7,8 of the first submission, and discredit the produced data set, to my opinion. Consequently, I don't think the intercomparisons provided in this manuscript are relevant as long as these biases are unexplained and uncorrected.

Specific comments

Once the biases have been corrected, the authors may also take the following specific comments into account for a revised submission:

1) Reference data sets

There is not reference provided for ESA WV_cci CDR-2 and it is indicated that this is a beta-version data record. In this case, I recommend to discard it from this study until a dedicated assessment report or publication for the data set is available. Reviewer #2 made a similar comment. The intercomparison would be more relevant based on ERA5 and RSS data which are well-established data sets.

2) the impact of time and space sampling and row anomaly are relevant to discuss in the manuscript but the details of the study would better fit into a supplemental material.

3) On the stability requirements (comments to the answer from the first review)

You are right that the GCOS-112 report that I quoted refers to radiosonde or profile measurements. However, since TCWV is the integral of profile data, the same requirements may be applied also to TCWV. This feeling is actually corroborated by the URD (2021) report that you are mentioning. Table 3.1 from this report gives indeed a 0.3%/decade stability requirement in TCWV based on 2 reference documents: GCOS (2016) and G-VAP (2013). The 1%/decade requirement which you cite from this report actually comes from a survey study conducted within the scientific community, but based on 38 answers only. I'm not sure to which number one should give more credit. At least, both numbers may be cited and properly acknowledged.

URD (2021) User Requirements Document of the ESA CCI WV (https://climate.esa.int/media/documents/Water_Vapour_cci_D1.1_URD_v3.0.pdf).

4) Linear regression method

In the first submission you used OLS and ODR without justification. You decided to remove OLS and replaced it with PWLF but still without justification and no details or reference citation (and you define the acronym as "piecewise linear regression", but I guess the F is for fitting). I guess your motivation for including PWLF is to account for the non-linear behaviour of the scatter plots in some cases (e.g. Fig. 9c) but you don't actually comment/interpret the change in the slopes. What is the physical explanation behind? I suspect that it just reveals the huge positive biases in the OMI TCWV data set over land. Moreover, comparing one ODR slope and two PWLF slopes does not make much sense (you are comparing the parameters of two different models). I also doubt that the PWLF model is relevant in some cases: Fig. 7, 9a, 10a where the break points are found at very small x values (even negative in Fig. 10a, although I cannot understand how this is possible).

5) Trend estimation

L350: The estimation method is said to be OLS and the following citations are quoted, but the details cannot be found actually:

Danielczok and Schröder (2017) is not available online

Beirle et al. (2018) does not explain how the trends are computed

Saunders et al. (2010) cited in Beirle et al. (2018) is not available online

Grossi (2017) cited in Beirle et al. (2018) used "a simple linear least-square regression analysis" and applied an augmented Dickey-Fuller (ADF) test, but this procedure is flawed. The ADF test only says if the series is stationary or not (null hypothesis). In order to take the serial correlation into account and derive properly scaled standard errors, a generalized least-square regression must be used instead.

6) Specification of the relative errors for the ODR

Following a comment both from myself and the 2nd reviewer, some justification of the specification has been added. However, they are still debatable. For SSMI, you cite Mears (2015), but I cannot find the numbers you are giving: 2-4 kg/m2 at 60 kg/m2. In their Table 4, the GPS-satellite difference reads 2.0-2.7 for 60 kg/m2, and this numbers obviously include errors from both GPS and satellite. For the other data sets, your choices are still very arbitrary. A comparison of the used data sets to a common ground truth (e.g. GNSS or radiosondes) may help to set proper values (it should be possible to find some in published work).

7) Correlation coefficients

Your answer to my comment on the correlation coefficients:

"With regard to the correlation coefficient, we cannot fully agree, as it includes spatial variation in addition to temporal variation: namely, if we reverse the latitudes, we only obtain a correlation of R=0.63 for RSS and R=0.45 for ERA5 over land."

I cannot undestand why the latitude reversal has an impact. Once the data are paired, the correlation coefficient does not depend on their sorting.

My comment was to say that if x and y include the seasonal signal (which is the dominant temporal variation for monthly time series), the correlation coefficient will always be very close to one. For monthly data, it is more relevant to compute the correlation coefficient of anomalies.

As an alternative, you may indicate the coefficient of determination ($R^2$) or better the adjusted $R^2$, as a measure of the goodness of fit.

---

## Referee Report (RR2)

**A 16-year global climate data record of total column water vapour generated from OMI observations in the visible blue spectral range**

Christian Borger, Steffen Beirle, and Thomas Wagner

**1. General Comments**

This paper assesses a long term record (2005-2020) of monthly mean total column water vapour (TCWV) from the Ozone Monitoring Instrument (OMI) on board NASAs Aura platform. The authors describe adaptations to an existing algorithm used for ESAs TROPOMI instrument, which is seen as a successor to OMI. This includes the rational for the how and why they switch to using earth shine spectra as the reference in the DOAS retrieval setup. This study goes on to present results from an inter-comparison of TCWV against two addition remote sensing products from RSS and ESA and ERA5 reanalysis.

Since my previous review, all comments have been adequately addressed and the recommend it for publication. There is one very minor technical correction I would ask the authors to change. The combined MW and NIR TCWV from ESA WV_cci is being referred to as 'COMBI', could this be changed to CM SAF/WV_cci or the text altered to say that this study refers to it as 'COMBI'. This way of defining the product has only really been settled on recently and I would not expect the authors to be aware.

---

## Referee Report (RR3)

Third review of manuscript "essd-2021-319-manuscript-version4.pdf" entitled "A 16-year global climate data record of total column water vapour generated from OMI observations in the visible blue spectral range" by Christian Borger, Steffen Beirle, and Thomas Wagner.

General comments

The data set described in this work clearly suffers from a major limitation, the large moist biases over land and near the coastlines. This has already been emphasized in the previous review reports but has not yet been solved and no directions to solve it are proposed in the manuscript. I do not see much utility of this data set as long as the intrinsic causes of these biases are not eliminated.

However, solving the moist bias problem may require substantial work on the retrieval method. If this problem is properly acknowledged and the directions to solve it are discussed (i.e. not just as a list of "possible" error sources), then the publication of an "interim" data set may be considered, but this data set should be limited to quality-assured data only (e.g. based on the effective clearing "masks").

Specific comments

1) Intercomparisons taking masks into account

What are the criteria and thresholds used to create the "common mask" of the ESA data set?

Illustrate on a map which grid points are filtered out by the threshold of 100 measurements of the "dynamic" monthly flag described in Appendix B. How does it connect to the percentage of observations shown in Fig. B3?

2) Reference data sets

The ESA "COMBI" product is still described as a beta-version data record with no validation results published. As such, it should not be considered as a "reference data set". The comparisons to this data set belong rather to a supplemental information section than to the main text.

What is the accuracy and stability of the IGRA-2 data set included in the new manuscript? It is well known that radiosonde data suffer from various biases in inhomogeneities and should not be used for trend analysis.

3) Sampling errors

Although the impact of time sampling is a relevant question, it is not currently a major limitation of the data set (positive biases). It should just be summarised with a few numbers in the text with the details (6 figures) provided as supplemental material.

It is not clear if the TCWV data from the different reference data sets (RSS, ERA5, etc.) are time-matched with the OMI data or if monthly mean products are used. Should be clarified.

4) Linear regression method

The use of two different regression methods (ODR and PWLR) is still not justified. What are the motivations?

Moreover, since both methods use different linear models and assumptions on the errors, it is difficult to compare their results.

In addition, the estimation of the break point $x_0$ in the PWLR is an ill-posed problem, how is it handled?

The PWLR results where x0 is very small are not relevant (e.g. Fig B4). Instead use either a fixed x0 (e.g. 25 kg/m2) or just use a model without break (in this case ODR is preferable).

Standard errors on the estimated slopes should be considered when results are compared.

5) Trend estimation

I am afraid the GLS method introduced in the new manuscript is not correctly implemented. Firstly, identifying the AR(p) order from the ACF is not robust. There is extensive literature on this subject.

The previously cited unavailable references have been removed, but one of the new reference is not available either (Prais and Winsten, 1954).

Secondly, the cited literature deals only with AR(1) models, not AR(3).

What is the formulation of the transformation matrix P in the case of AR(3)?

Why are the new trend estimates are very different from those in the previous manuscript?

Standard errors should be given on the estimated trends, and statistical significance should be quantified by a p-value or a significance level.

The comparisons to the COMBI and the IGRA-2 data sets show large trends, which do not fulfil the stability requirements. These results should be preferably removed or at least moved to a supplemental material and interpreted with caution.

6) Correlation coefficients

My suggestion to use the of determination ($R^2$) rather than the correlation coefficient applied of course to TCWV anomalies (not to TCWV) since both $R^2$ and R suffer from the same limitation (being close to one when the seasonal signal is included).

---

## Author Response (AR2)

Dear Editors,

We have revised the manuscript according to the recommendations of the reviewers. Below we will list the most important changes compared to the manuscript version of the first revised submission (essd-2021-319-manuscript-version4.pdf). A pdf manuscript created with LatexDiff is attached to this "Author's Response".

We have revised the stability analysis and now consider the effect of temporal autocorrelation when calculating stability trends, giving a detailed description of our procedure (Section 6). The new stability results are now also in line with the requirements of the latest GCOS report.

Furthermore, we have also now made comparisons with in situ radiosonde measurements from the IGRA2 data set to complement the results of the intercomparisons to the global satellite and reanalysis reference data (Section 5). These new comparisons to in situ measurements support the previously obtained results from the global datasets.

In addition, we have now introduced flags to indicate less trustworthy data or data with greater uncertainty. In detail, these are a flag for coastlines and a flag for the number of measurements used to calculate the monthly means of the grid cell (Appendix B).

We thank the referee for the second review of the manuscript. Below we reply to the issues raised by the referee, where

blue repeats the reviewer's comments,

black is used for our reply,

*and green italics is used for modified text and new text added to the manuscript.*

2nd review of manuscript "essd-2021-319-manuscript-version4.pdf" entitled "A 16-year global climate data record of total column water vapour generated from OMI observations in the visible blue spectral range" by Christian Borger, Steffen Beirle, and Thomas Wagner.

General comments

I acknowledge for the additional work done by the authors to address the questions of time and space sampling and clear-sky bias. However, I cannot find the answers to my main criticism and recommendations, which are still pending. Let me recall them:

"the conclusions sound far too optimistic to me, given the poor agreement found between the OMI data and the validation data. Especially, the large positive biases over land and near the coastlines in the tropics are striking and not sufficiently commented or explained."

"I recommend first a more insightful analysis of the error sources, especially over land and, if possible, the elaboration of an improved version of the data set, and second, a more comprehensive discussion of the validation results in a revised version of the manuscript.

The biases in the proposed data set are striking, see the reddish patterns in Figs. 3,4,5,6,7,8 of the first submission, and discredit the produced data set, to my opinion. Consequently, I don't think the intercomparisons provided in this manuscript are relevant as long as these biases are unexplained and uncorrected.

We cannot fully follow the reviewer, as we explicitly address the (possible) sources of error of the positive deviations over the tropical land masses (albedo too low, erroneous cloud information, low number of observations) as well as along the coastlines (poor spatial resolution of the albedo map). These errors are widely known in the (DOAS) satellite community and a detailed error analysis of TCWV retrievals in the visible blue spectral range is also available in Borger et al. (2020).

Regarding a bias correction, we believe that this would also only be an "artificial" intervention in the data. Instead, we propose to add two more variables to the dataset:

- First, a flag that distinguishes between ocean, land and coastline, so that interested users can filter these problematic regions.
- Second, the number of measurements per spatial bin used to calculate the monthly mean. This allows regions with a small sample size and thus lower trustworthiness to be filtered as well. This corresponds to the effect of the "common mask" of the ESA data set.

We would also like to note that for climate applications (e.g. trend analyses) biases are not the decisive point as long as they are clearly named in the paper and understood in principle. The main advantages of our data set are its temporal stability and global coverage, not the perfect measurement accuracy of individual measurements.

Regarding the points above, we have added the following text to Appendix B (in which the ESA/COMBI common mask was introduced):

*Appendix B: Intercomparisons taking into account masks and flags*

[revised manuscript text omitted]

Specific comments

Once the biases have been corrected, the authors may also take the following specific comments into account for a revised submission:

1) Reference data sets
There is not reference provided for ESA WV_cci CDR-2 and it is indicated that this is a beta-version data record. In this case, I recommend to discard it from this study until a dedicated assessment report or publication for the data set is available. Reviewer #2 made a similar comment.

The intercomparison would be more relevant based on ERA5 and RSS data which are well-established data sets.

The ESA WV_CCI dataset is another independent TCWV dataset that can provide new insights, especially over land, as its data is not assimilated into ERA5. Thus, we believe that this dataset is relevant for the comparative studies.

2) the impact of time and space sampling and row anomaly are relevant to discuss in the manuscript but the details of the study would better fit into a supplemental material.

Reviewer #2 explicitly requested such an investigation, which is why we want to refrain from removing it from the manuscript. However, if the referee insists, we can move this section to the appendix.

3) On the stability requirements (comments to the answer from the first review)

You are right that the GCOS-112 report that I quoted refers to radiosonde or profile measurements. However, since TCWV is the integral of profile data, the same requirements may be applied also to TCWV. This feeling is actually corroborated by the URD (2021) report that you are mentioning. Table 3.1 from this report gives indeed a 0.3%/decade stability requirement in TCWV based on 2 reference documents: GCOS (2016) and G-VAP (2013). The 1%/decade requirement which you cite from this report actually comes from a survey study conducted within the scientific community, but based on 38 answers only. I'm not sure to which number one should give more credit. At least, both numbers may be cited and properly acknowledged.

URD (2021) User Requirements Document of the ESA CCI WV (https://climate.esa.int/media/documents/Water_Vapour_cci_D1.1_URD_v3.0.pdf).

In the revised version we address the two references mentioned regarding stability. In addition, recently in the latest GCOS report, further stability criteria have now been explicitly set for TCWV. We will also address these together with the issues raised under point (5).

4) Linear regression method

In the first submission you used OLS and ODR without justification. You decided to remove OLS and replaced it with PWLF but still without justification and no details or reference citation (and you define the acronym as "piecewise linear regression", but I guess the F is for fitting). I guess your motivation for including PWLF is to account for the non-linear behaviour of the scatter plots in some cases (e.g. Fig. 9c) but you don't actually comment/interpret the change in the slopes. What is the physical explanation behind? I suspect that it just reveals the huge positive biases in the OMI TCWV data set over land. Moreover, comparing one ODR slope and two PWLF slopes does not make much sense (you are comparing the parameters of two different models). I also doubt that the PWLF model is relevant in some cases: Fig. 7, 9a, 10a where the break points are found at very small x values (even negative in Fig. 10a, although I cannot understand how this is possible).

We have now changed the name of the regression using "piecewise linear regression" to PWLR. The idea behind this is to automatically determine the influence of the different TCWV regimes or the limits of these regimes for data over land, so that it can be determined up to which TCWV the data set is trustworthy. For consistency reasons, this was therefore also applied for ocean data.

To clarify which function is fitted in the PWLR, we added the following text to the manuscript:

*For the PWLR, a function of the form*

$$f(x) = \begin{cases} a_0 \cdot x + b_0 & x < x_0 \\ a_1 \cdot x + b_1 & x > x_0 \end{cases}$$

*is assumed, whereby the function parameters (including $x_0$) are determined via a non-linear least-squares fit.*

5) Trend estimation

L350: The estimation method is said to be OLS and the following citations are quoted, but the details cannot be found actually:

Danielczok and Schröder (2017) is not available online

Beirle et al. (2018) does not explain how the trends are computed

Saunders et al. (2010) cited in Beirle et al. (2018) is not available online

Grossi (2017) cited in Beirle et al. (2018) used "a simple linear least-square regression analysis" and applied an augmented Dickey-Fuller (ADF) test, but this procedure is flawed. The ADF test only says if the series is stationary or not (null hypothesis). In order to take the serial correlation into account and derive properly scaled standard errors, a generalized least-square regression must be used instead.

Thank you very much for this hint and for pointing out that some documents are no longer available online! We have carried out the stability analysis again and taken into account the influence of the autocorrelation by means of a Prais-Winsten transformation. We selected the lag of the AR model so that the ACF is smaller than 0.5 at this lag, which is typically the case after lag=2 or 3. Thus, the stability values have changed and are now also in line with the latest GCOS criteria.

The respective section in the manuscript has been revised as follows:

*In addition to a good agreement to existing reference data sets, the temporal stability is an important property of a climate data record. As the COMBI data set only covers the time range up to December 2017, we focus on the comparison to the RSS SSM/I and ERA5 data sets as these two cover the complete time range of OMI TCWV data set. For the sake of completeness, however, we also show the results for COMBI.*

*To assess the stability of the OMI TCWV data set, first the global mean relative deviation $\langle \epsilon \rangle$ is derived for every time step:*

$$\langle \epsilon \rangle = \frac{\langle OMI - TCWV_{ref} \rangle}{\langle TCWV_{ref} \rangle}$$

*For the calculation of global means only data points or grid cells are taken into account for which for every time step data from the OMI TCWV and reference data set are available. In the case of the COMBI data set a "common mask" has been provided (see also Fig. B1).*

*Then, temporal linear trends of these deviations are calculated using a generalized least-squares (GLS) regression for the fit function:*

$$Y_t = m + b \cdot X_t = \boldsymbol{M}_t x + N_t$$

*with the intercept m, the trend b, the increasing time index Xt (in months), which can all be summarised in a matrix Mt. The term Nt stands for the fit residuals with respect to the time series. To account for the temporal autocorrelation of the fit residuals Nt of the GLS, the Prais-Winsten transformation (Prais and Winsten, 1954) is used assuming that the residuals follow an autoregressive process. For this purpose, the autocorrelation (ACF) is estimated using the Gaussian-kernel-based cross-correlation function algorithm, as described in Rehfeld et al. (2011). When examining the ACF of the fit residuals, it was found that the ACF is smaller than 0.5 only from a lag of 3. Thus, the GLS system is transformed by means of a 3rd order autoregressive process using the transformation matrix P:*

$$PY_t = P(\boldsymbol{M}_t x + N_t) = \boldsymbol{M}_t' x + \varepsilon_t$$

[revised manuscript text omitted]

6) Specification of the relative errors for the ODR

Following a comment both from myself and the 2 nd reviewer, some justification of the specification has been added. However, they are still debatable. For SSMI, you cite Mears (2015), but I cannot find the numbers you are giving: 2-4 kg/m2 at 60 kg/m2. In their Table 4, the GPS-satellite difference reads 2.0-2.7 for 60 kg/m2, and this numbers obviously include errors from both GPS and satellite. For the other data sets, your choices are still very arbitrary. A comparison of the used data sets to a common ground truth (e.g. GNSS or radiosondes) may help to set proper values (it should be possible to find some in published work).

In the revised manuscript we also mention the paper by Wentz (1997) as a reference for the relative error. In this study, a theoretical error estimate for SSM/I determined a mean error of 1.27mm. With a typical TCWV of 25-30mm, this corresponds approximately to the 5% we assumed.

The manuscript was revised as follows:

*In a comprehensive uncertainty analysis, Wentz (1997) determined a typical error of 1.22kg/m2 for SSM/I observations. Mears et al. (2015) found that the uncertainty of daily microwave TCWV observations for TCWV = 10kg/m2 was around 1kg/m2 and for TCWV = 60kg/m2 around 2-4kg/m2 with respect to GNSS measurements. Hence, we assume that the uncertainty of the RSS data set is 5% or at least 1kg/m2.*

Moreover, we decided to conduct an additional comparison with radiosonde data from IGRA2.

**5. Intercomparison to IGRA2 radiosonde observations**

[revised manuscript text omitted]

7) Correlation coefficients

Your answer to my comment on the correlation coefficients:

"With regard to the correlation coefficient, we cannot fully agree, as it includes spatial variation in addition to temporal variation: namely, if we reverse the latitudes, we only obtain a correlation of R=0.63 for RSS and R=0.45 for ERA5 over land."

I cannot undestand why the latitude reversal has an impact. Once the data are paired, the correlation coefficient does not depend on their sorting.

My comment was to say that if x and y include the seasonal signal (which is the dominant temporal variation for monthly time series), the correlation coefficient will always be very close to one. For monthly data, it is more relevant to compute the correlation coefficient of anomalies.

As an alternative, you may indicate the coefficient of determination ($R^2$) or better the adjusted $R^2$, as a measure of the goodness of fit.

We thank the reviewer for clarifying his comment and apologise for the misunderstanding. In the revised manuscript in all relevant figures, the correlation coefficient is replaced by the coefficient of determination for ODR fit (the adjusted $R^2$ is not necessary due to the large sample size).

We would like to thank the referee for reviewing our manuscript. Below we reply to the issues raised by the referee, where

blue repeats the reviewer's comments,

black is used for our reply,

*and green italics is used for modified text and new text added to the manuscript.*

A 16-year global climate data record of total column water vapour generated from OMI observations in the visible blue spectral range

Christian Borger, Steffen Beirle, and Thomas Wagner

1. General Comments

This paper assesses a long term record (2005-2020) of monthly mean total column water vapour (TCWV) from the Ozone Monitoring Instrument (OMI) on board NASAs Aura platform. The authors describe adaptations to an existing algorithm used for ESAs TROPOMI instrument, which is seen as a successor to OMI. This includes the rational for the how and why they switch to using earth shine spectra as the reference in the DOAS retrieval setup. This study goes on to present results from an inter-comparison of TCWV against two addition remote sensing products from RSS and ESA and ERA5 reanalysis.

Since my previous review, all comments have been adequately addressed and the recommend it for publication. There is one very minor technical correction I would ask the authors to change. The combined MW and NIR TCWV from ESA WV_cci is being referred to as 'COMBI', could this be changed to CM SAF/WV_cci or the text altered to say that this study refers to it as 'COMBI'. This way of defining the product has only really been settled on recently and I would not expect the authors to be aware.

We would like to thank the reviewer for the positive feedback. In the revised manuscript, the suggested changes regarding the name of the ESA product are applied, i.e. the product will now be called "COMBI".

[revised manuscript text omitted]

---

## Author Response (AR3)

Dear Editors,

We have revised the manuscript according to the recommendations of the reviewer. Below we will list the most important changes compared to the manuscript version of the second revised submission (essd-2021-319-manuscript-version5.pdf). A pdf manuscript created with LatexDiff is attached to this "Author's Response".

We have revised the stability analysis again and now estimate the lag p of the AR(p) model via the partial autocorrelation function (PACF), resulting in an AR(2) model being used over land and an AR(3) model over ocean surface. Also, we now indicate the fit errors of the slope/trend of the regression.

Moreover, we have optimised the colorscale of the difference maps where necessary and now give a wider range (+-15kg/m2) so that oversaturations no longer occur in the maps. For the sake of clarity, we have also removed the PWLR fits from the scatterplots for data over ocean and explained why we additionally use the PWLR method over land.

Furthermore, we noticed a small error when resampling the COMBI data set, which caused the grid cells to be shifted by 0.5° and thus caused the large deviations along the west coasts. By correcting this error, these artefacts have now largely disappeared.

In addition, the manuscript was restructured a bit, so that, for example, the stability analysis with the IGRA2 data set was moved to an Appendix section.

Finally, we also extended the data set with 2 additional variables (namely "surface_type_flag" and the number of observations per grid cell "num_obs") and checked for CF compliance (using https://github.com/cedadev/cf-checker). The updated dataset is available at doi:10.5281/zenodo.7973889

We thank the referee for the third review of the manuscript. Below we reply to the issues raised by the referee, where

blue repeats the reviewer's comments,

black is used for our reply,

*and green italics is used for modified text and new text added to the manuscript.*

Third review of manuscript "essd-2021-319-manuscript-version4.pdf" entitled "A 16-year global climate data record of total column water vapour generated from OMI observations in the visible blue spectral range" by Christian Borger, Steffen Beirle, and Thomas Wagner.

General comments
The data set described in this work clearly suffers from a major limitation, the large moist biases over land and near the coastlines. This has already been emphasized in the previous review reports but has not yet been solved and no directions to solve it are proposed in the manuscript. I do not see much utility of this data set as long as the intrinsic causes of these biases are not eliminated.
However, solving the moist bias problem may require substantial work on the retrieval method. If this problem is properly acknowledged and the directions to solve it are discussed (i.e. not just as a list of "possible" error sources), then the publication of an "interim" data set may be considered, but this dataset should be limited to quality-assured data only (e.g. based on the effective clearing "masks").
While we agree with the reviewer that the high deviations in the tropical regions are a major limitation of the data set, we disagree on the statement that the data set has not much utility due to these biases, especially taking into account that other satellite data sets also show high deviations of a similar magnitude as the OMI TCWV data set in the tropical regions with high VCDs. For instance, TCWV data sets from AIRS or IASI underestimate high VCDs (> 60kg/m2) by about 20%, i.e. more than 10-15kg/m2 in tropical regions (see Roman et al. (2016), Trent et al. (2019) and the validation report in Schröder et al. (2023)). Depending on the scientific question, the user can simply use the data, which fit his/her accuracy requirement and skip data with higher uncertainties. Here we also want to point out that the major advantages of the OMI TCWV data set are that it was generated from only one single instrument and that it provides global coverage over land and ocean surface. This makes the data set very valuable for trend analyses, for which possible local biases are not relevant, but potential "jumps" that can occur when time series of different instruments are blended/merged together.
The uncertainties of the TCWV retrieval (especially in the tropical regions) are mainly due to the uncertainties in the input data, as already mentioned. Consequently, one way to address these errors would be to develop an independent albedo and cloud product, but this is far beyond the scope of this paper. In particular, the L1 data are currently being reprocessed by the OMI team and considering the soon demise of OMI (probably in 1-2 years), such an algorithm development would not be worthwhile at the moment.
While a simple bias correction with respect to ERA5 could be applied to the OMI TCWV data set, we believe this would almost nullify the purpose of the data set (then one could simply use ERA5). Therefore, we believe that open communication of the weaknesses (and strengths) of the data set is a more useful contribution to the existing TCWV data sets than adapting the data set to the reference data. Hence, as mentioned above, we want to leave it up to the user to decide which values to use and therefore have decided to introduce flags for regions with high uncertainties (e.g. coastlines and grid cells with few observations).
Regarding the overestimations along coastlines compared to the COMBI dataset, we found that we made a small error in the resampling of the COMBI dataset, so that the grid cells were shifted by 0.5°. This error has been corrected and the large biases along the coasts (e.g. west coast of America) have largely disappeared.
If it is important to the reviewer, we are also willing to call the data set an "interim" climate data record (ICDR) in view of the current uncertainties, but we would prefer to keep the name as it

is, because we see not much value in that change (every data set is subject to possible future improvements).

Specific comments
1) Intercomparisons taking masks into account
What are the criteria and thresholds used to create the "common mask" of the ESA data set?
As mentioned in Sect. 4.1 and according to the validation report of the ESA data set, the common mask includes only the grid cells, for which "valid TCWV values […] over all time steps in the time period July 2002 to March 2016" are available.
We added the following text to Appendix B in the revised manuscript:
*The "common mask" only considers grid cells for which valid TCWV values were available over all time steps in the COMBI data set in the time period July 2002 to March 2016 (Schröder et al., 2023).*

Illustrate on a map which grid points are filtered out by the threshold of 100 measurements of the "dynamic" monthly flag described in Appendix B. How does it connect to the percentage of observations shown in Fig. B3?
Since the mask from the OMI data set is not static but rather "dynamic" (i.e. it changes from month to month), the corresponding distribution of the grid cells that fulfil this criterion also changes with the annual cycle. As an example, the distributions for March, June, September, and December 2005 are shown in the figure below.

[Figure]

The figure shows that in the subtropics, which are virtually cloud-free, a high number of satellite observations are available throughout the year. Accordingly, the fractional coverage in Fig. B3 is close to 100%. The low values in the mid to high latitudes in Fig. B3 are due to the annual variation of the solar zenith angle and the seasonal degree of cloud cover as well as ice cover. Within the tropics, the frequently occurring high cloud cover is mainly responsible for the low number of measurement data. As a result, the values in Fig. B3 for mid- and high latitudes as well as for the tropics do not achieve 100% temporal coverage.

2) Reference data sets
The ESA "COMBI" product is still described as a beta-version data record with no validation
results published. As such, it should not be considered as a "reference data set". The
comparisons to this data set belong rather to a supplemental information section than to the
main text.
Meanwhile, the ESA COMBI data set has been officially published (Schröder et al., 2023) and
is not a beta-product anymore. Moreover, an extensive validation study of this climate data
set has been performed including intercomparisons to global reference data sets (e.g. ERA5),
satellite products (AIRS, GOME-Evolution), GPS-measurements from SuomiNet, and GRUAN
radiosonde observations (more details in the validation report of Schröder et al., 2023).
Furthermore, this data set is one of the few (satellite) measurement data sets that provide
global coverage over ocean and land surface. As such, we would keep the intercomparison
with respect to this product in the main body of the manuscript.

What is the accuracy and stability of the IGRA-2 data set included in the new manuscript? It
is well known that radiosonde data suffer from various biases in inhomogeneities and should
not be used for trend analysis.
We agree with the reviewer that radiosonde data may be affected by biases and
inhomogeneities. Regarding the IGRA2 data, to our knowledge so far no explicit uncertainty
estimates for water vapour measurements have been conducted. However, the IGRA2
humidity measurements are subject to rigorous quality control (Durre et al., 2018) and the
completeness of the IGRA humidity observations has also been checked by Ferreira et al.
(2019). Hence, we assume that the uncertainties of the TCWV measurements of the IGRA2
data set should be within the typical values for radiosonde measurements of about 5% (e.g.
Dirksen et al., 2014), so that the potential uncertainties and biases play a minor role compared
to those of satellite measurements.
With respect to the stability analysis using IGRA2, we have added this analysis mainly for the
sake of completeness. In the revised version we have moved this to the supplement.

3) Sampling errors
Although the impact of time sampling is a relevant question, it is not currently a major limitation
of the data set (positive biases). It should just be summarised with a few numbers in the text
with the details (6 figures) provided as supplemental material.
Reviewer 2 explicitly insisted that such an analysis of sampling errors should be conducted.
Thus, we would like to keep this analysis in the main body of the manuscript.

It is not clear if the TCWV data from the different reference data sets (RSS, ERA5, etc.) are
timematched with the OMI data or if monthly mean products are used. Should be clarified.
All reference data for the comparison are monthly mean products. In the revised manuscript,
the following text was added to section 4:
*To evaluate the overall quality of the OMI TCWV data set, we conducted an intercomparison
study to various reference data sets of monthly mean TCWV products. For this purpose, we
use …*

4) Linear regression method
The use of two different regression methods (ODR and PWLR) is still not justified. What are
the motivations?
Moreover, since both methods use different linear models and assumptions on the errors, it is
difficult to compare their results.
When first looking at the scatter plots of the comparisons over land, it was noticed that for low
TCWV values the data is close to the 1-1 line, but for higher values it deviates from it
significantly. Thus, the results of the ODR are not representative for all TCWV regimes (i.e.
low and high TCWV values). In order to address this problem of the 2 regimes, the PWLR
method was additionally applied to have an alternative solution that can provide an objective
quantification of the quality/goodness of the OMI TCWV data set for both TCWV regimes.

We added the following text to the beginning of Section 4:
*First comparisons with the reference data over land indicated that the OMI data set shows different levels of agreement for low and high VCD values, with high deviations being particularly prominent for high TCWV values. To be able to estimate the goodness for low and high TCWV, a piecewise linear regression (PWLR) is additionally performed for data over land.*

In addition, the estimation of the break point x0 in the PWLR is an ill-posed problem, how is it handled?
For the PWLR method, all function parameters (slopes, intercepts and the breakpoint) are estimated using a non-linear least-squares fit. We also investigated to what extent the results change depending on the initial values and could not find any dependence. Typically, the error values for the breakpoint are in the range of 0.03-0.04 and for the slope in the range of 0.0004-0.0006, thus for both variables about four orders of magnitude smaller than their values (25kg/m2 and 1.0, respectively) and thus negligible.

The PWLR results where x0 is very small are not relevant (e.g. Fig B4). Instead use either a fixed x0 (e.g. 25 kg/m2) or just use a model without break (in this case ODR is preferable).
The non-physical results of the PWLR method for comparisons over ocean indicate that, in contrast to the comparisons over land, the quality/goodness of the OMI data set does not depend on the TCWV. For the sake of completeness, we had decided to show these fit results as well. In the revised manuscript, however, for a better overview of the figures we have removed the PWLR fit results from the scatterplots of the comparisons over ocean surface.

Standard errors on the estimated slopes should be considered when results are compared.
The standard error of the ODR and PWLR slopes is typically in the order of 1e-4 and reaches a maximum value of 0.0007 for the intercomparison to the COMBI data set over land. Thus, the standard errors are negligible compared to the estimated slope values (typically around 1.0), so we would refrain from mentioning them in the revised manuscript.

5) Trend estimation
I am afraid the GLS method introduced in the new manuscript is not correctly implemented. Firstly, identifying the AR(p) order from the ACF is not robust. There is extensive literature on this subject.
We would like to thank the reviewer for the hint regarding the calculation of the AR order and follow his recommendation. We now determine the AR(p)-order using the partial autocorrelation function (PACF) and investigate at which lag p the values of the PACF are within a confidence interval. Here, the boundaries of the confidence interval are estimated by the Z-score for a significance level of 95% divided by the square root of the number of length of the series: Z / sqrt(N). It turns out that over land p=2 and ocean (or both surfaces together) p=3 is the optimal choice (using a Z-score at a significance level of 95%). The corresponding new stability analyses will be incorporated in the revised manuscript and the following text was added:
*For the estimation of the order of the AR model, we use the partial autocorrelation function (PACF) and investigate after which lag all values of the PACF lie within a confidence interval $\pm\delta$. Assuming that the PACF values for high lags follow a white noise, the confidence interval is defined by the Z-score (in our case of a significance level of 95%) and the length of the time series L according to the following formula (Box et al., 2015): $\delta = \frac{Z}{\sqrt{L}}$*

The previously cited unavailable references have been removed, but one of the new reference is not available either (Prais and Winsten, 1954).
Thank you very much for this hint! We have removed the link and added Greene (2019) as a further reference.

Secondly, the cited literature deals only with AR(1) models, not AR(3). What is the formulation of the transformation matrix P in the case of AR(3)?

Analogous to the AR(1) model, the transformation matrix for AR(3) can be created as follows:

$$P_{AR(3)} = \begin{pmatrix} \alpha & 0 & & & & \\ -\varphi_1 & 1 & 0 & & & \\ -\varphi_2 & -\varphi_1 & 1 & 0 & & \\ -\varphi_3 & -\varphi_2 & -\varphi_1 & 1 & 0 & \\ 0 & -\varphi_3 & -\varphi_2 & -\varphi_1 & 1 & \\ & \ddots & \ddots & \ddots & \ddots & \ddots \end{pmatrix}$$

where the AR coefficients $\varphi$ were determined via the Yule-Walker equations and $\alpha = \sqrt{1 - \sum_i \rho_i \varphi_i}$ with $\rho_i$ being the coefficients of the autocorrelation function.

Why are the new trend estimates are very different from those in the previous manuscript?

The presence of the autocorrelation violates the Gauss-Markov conditions for linear regression, so the former fit results did not represent the best linear unbiased estimator and were not statistically efficient (i.e. did not have the smallest possible variance). As a consequence, the former fit results also deviated from the "truth". By taking into account the autocorrelation, this issue is resolved.

Standard errors should be given on the estimated trends, and statistical significance should be quantified by a p-value or a significance level.

In the revised manuscript we have added the standard errors of the fit results, which are of the order of about 1-2% per decade. In this way, we also believe that further tests of statistical significance are not absolutely necessary. In addition, standard test procedures (Z-test, t-test) would only be of limited use, as these tests check whether the results obtained are different from 0. In this way, however, one cannot conclude from a null hypothesis that has not been rejected that the stability trend is equal to 0.

Instead, a so-called equivalence test is more useful (see Dixon and Pechmann, 2005). In our case, one must therefore define an equivalence region around 0% per decade and then conduct two one-sided t- or Z-tests. We applied this procedure and used the "goal" value of GCOS as the equivalence region (i.e. 0.1% per decade). It turns out that for all stability analyses the null hypothesis of the equivalence test cannot be refuted, as the uncertainties are significantly larger than the fit results for the stability trend.

The comparisons to the COMBI and the IGRA-2 data sets show large trends, which do not fulfil the stability requirements. These results should be preferably removed or at least moved to a supplemental material and interpreted with caution.

We moved the IGRA2 analysis to the supplements.

6) Correlation coefficients

My suggestion to use the of determination (R^2) rather than the correlation coefficient applied of course to TCWV anomalies (not to TCWV) since both R^2 and R suffer from the same limitation (being close to one when the seasonal signal is included).

Removing the seasonal cycle leads to the following R^2 values:
- RSS: 0.50
- ERA5:  0.49 over ocean and 0.40 over land
- COMBI:  0.45 over ocean and 0.32 over land
All these values are mentioned in the revised manuscript.

[revised manuscript text omitted]